# CONNECTING DOMAINS AND CONTRASTING SAMPLES: A LADDER FOR DOMAIN GENERALIZATION

## ABSTRACT

Distribution shifts between training and testing datasets, contrary to classical machine learning assumptions, frequently occur in practice and impede model generalization performance. Studies on domain generalization (DG) thereby arise, aiming to predict the label on unseen target domain data by only using data from source domains. In the meanwhile, the contrastive learning (CL) technique, which prevails in self-supervised pre-training, can align different augmentation of samples to obtain invariant representation. It is intuitive to consider the class-separated representations learned in CL are able to improve domain generalization, while the reality is quite the opposite: people observe directly applying CL deteriorates the performance. We analyze the phenomenon with the CL theory and discover the lack of intra-class connectivity in the DG setting causes the deficiency. Thus we propose domain-connecting contrastive learning (DCCL) to enhance the conceptual connectivity across domains and obtain generalizable representations for DG. Specifically, more aggressive data augmentation and cross-domain positive samples are introduced into self-contrastive learning to improve intra-class connectivity. Furthermore, to better embed the unseen test domains, we propose model anchoring to exploit the intra-class connectivity in pre-trained representations and complement it with generative transformation loss. Extensive experiments on five standard DG benchmarks are provided. The results verify that DCCL outperforms state-of-the-art baselines even without domain supervision.

## 1 INTRODUCTION

Neural networks have achieved great progress in various vision applications, such as visual recognition (He et al., 2016), object detection (Tan et al., 2020), semantic segmentation (Cheng et al., 2021), pose estimation (Sun et al., 2019), etc. Despite the immense success, existing approaches for representation learning typically assume that training and testing data are independently sampled from the identical distribution. However, in real-world scenarios, this assumption does not necessarily hold. In image recognition, for example, distribution shifts w.r.t. geographic location (Beery et al., 2018) and image background (Fang et al., 2013) frequently occur and impede the generalization performance of models.

Accordingly, domain generalization (DG) (Gulrajani & Lopez-Paz, 2020) is widely studied to strengthen the transferability of deep learning models. Different from domain adaptation (DA) (You et al., 2019; Tzeng et al., 2017) where unlabeled or partially labeled data in target domains are available during training, in a DG task we can only resort to source domains. A natural idea for DG is to learn invariant representation across a variety of seen domains so as to benefit the classification of unobserved testing domain samples. As a powerful representation learning technique, contrastive learning (CL) (Chen et al., 2020) aims to obtain class-separated representations and has the potential for DG (Yao et al., 2022). In this paper, however, we observe that the widely deployed self-contrastive learning (SCL) (Chen et al., 2020; He et al., 2020; Grill et al., 2020), which aligns the augmentation of the same input, does not naturally fit the domain generalization setting: it implicitly assumes the capability to sample instances from the whole data distribution.

To bridge this gap, we propose domain-connecting contrastive learning (DCCL) to pursue transferable representations in DG, whose core insight comes from a novel understanding of CL attributing the success of CL to the intra-class representation connectivity (Wang et al., 2022b). Specif-

ically, we first suggest two direct approaches to improve *intra-class connectivity* (to be fully explained at the beginning of Section 2) within CL: (i) applying more aggressive data augmentation and (ii) expanding the scope of positive samples from self-augmented outputs to the augmentation of same-class samples across domains. In addition to the direct approaches, we have an interesting observation that the pre-trained models, unlike the learned maps, indeed possess the desired intra-class connectivity: the intra-class samples of the training domains and the testing domains are scattered but well-connected. The encouraging observation motivates us to anchor learned maps to the pre-trained model and further complement it with a generative transformation loss for stronger intra-class connectivity. As a visual illustration, Figure 1 demonstrates the embeddings learned by regular Empirical Risk Minimization (ERM) and by the proposed DCCL. ERM embeds the data in a more scattered distribution, and many samples in the central region cannot be distinguished; on the other hand, DCCL can well cluster and separate inter-class samples regardless of the domains. It verifies the effectiveness of our proposed DCCL on connecting domains.

Our contributions are summarized as follows:
(i) We analyze the failure of self-contrastive learning on DG and propose two effective strategies to improve intra-class connectivity within CL.
(ii) We propose to anchor learned maps to pre-trained models which possess the desired connectivity of training and testing domains. Generative transformation loss is further introduced to complement the alignment in between.

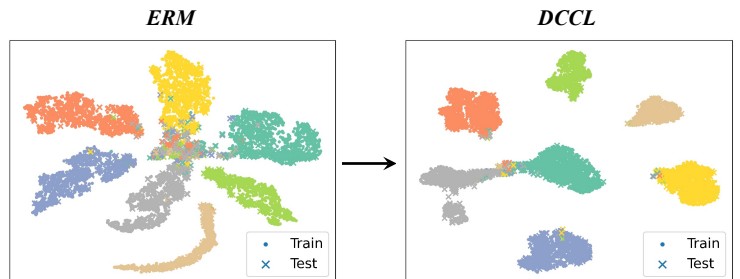

Figure 1: Visualization for ERM and DCCL on PACS. Intra-class points have the same colors, and two marker types differentiate the training and testing domains. Our proposed method better bridges the intra-class samples across domains than ERM.

(iii) We conduct extensive experiments on five real-world DG benchmarks with various settings, demonstrating the effectiveness and rationality of DCCL.

## 2 PRELIMINARIES

We first illustrate the core concept of the paper, *intra-class connectivity*. It refers to the intra-class data connectivity across different domains and resembles the connectivity in CL theory (Wang et al., 2022b), which depicts the preference that samples should not be isolated from other intra-class data of the same class [1]. In the remainder of this section, we introduce problem formulation and necessary preliminaries for contrastive learning in this section. A thorough review of related work on domain generalization and contrastive learning are deferred to Appendix B due to space limit.

### 2.1 DATA IN THE DOMAIN GENERALIZATION SETTING

Given $N$ observations (from $M$ domains), $\mathbf{X} = \{x_1, \ldots, x_N\} \subseteq \mathcal{X}$ is the collection of input features, $\mathbf{Y} = \{y_1, ..., y_N\} \subseteq \mathcal{Y}$ represents the prediction targets, and the whole dataset $D_s$ is represented as $\{(x_i^m, y_i^m)_{i=1}^{N_m}\}_{m=1}^{M}$, where $N_m$ is the number of samples ($\sum_{m=1}^{M} N_m = N$) in the domain $d_m$ and $x_i$ is re-indexed as $x_i^m$ accordingly.

The goal of this paper is to train a generalizable classification model from partial domains in $D_s$, which has satisfactory performance even on the unseen domains in evaluation. We also follow the specific settings in Cha et al. (2021; 2022); Chen et al. (2022) where only the feature vector $x_i \in \mathbf{X}$ and the label $y_i \in \mathbf{Y}$ are observable, while the domain identifier $d_m \in \mathbf{D}$ cannot be explicitly utilized due to the expensive cost.

---

[1]An intuitive graph-based measure to assess the intra-class connectivity of a given model is discussed in Appendix A.4.

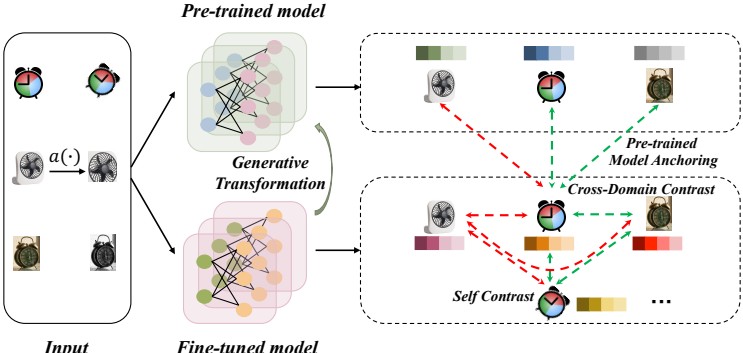

Figure 2: The overall framework of DCCL. The green dotted arrows indicate the two representations form a positive pair and the red ones connect the negative pairs. $a(\cdot)$ is an augmentation operation. Three key parts in DCCL are (i) cross-domain contrast to bridge the intra-class samples across domains; (ii) pre-trained model anchoring to further possess the intra-class connectivity; (iii) generative transformation to complement the pre-trained representation alignment.

## 2.2 CONTRASTIVE LEARNING

Contrastive Learning (CL) enforces the closeness of augmentation from the same input, compared to other inputs in the representation space. The main components of CL, as summarized in Chen et al. (2020); He et al. (2020), include: (i) data augmentation for contrastive views, (ii) a representation map $f$ as the data encoder: $\mathcal{X} \to \mathbb{R}^d$, (iii) projection head $h(\cdot)$ for expressive representation, and (iv) contrastive loss for optimization. Given an instance from $\mathbf{X}$, we draw a positive pair $x, x^+$ by applying a random data augmentation $a \sim \mathcal{A}$, where $\mathcal{A}$ is the pre-specified distribution of random data augmentation maps. As a contrastive concept to positive samples, a negative pool $\mathcal{N}_x$ is the set of augmented samples randomly drawn from the whole dataset $\mathbf{X}$. To ease the construction of the CL loss, we denote $p(x)$ as the distribution of $x$, $p(x, x^+)$ as the corresponding joint distribution of the positive pairs, and $p_n(x_i^-)$ ("n" is shorthand for "negative") as the distribution for $x_i^- \in \mathcal{N}_x$, which are all independent and identically distributed (i.i.d.). Let $z$ denote the normalized outputs of input feature $x$ through $f_h := (h \circ f)(\cdot)$. Consequently, $z^+ = f_h(x^+)$ is the positive embedding of $z = f_h(x)$, and $z_i^- = f_h(x_i^-)$ represents the embedding of the samples in the negative pool $\mathcal{N}_x$.

The most common form of the CL loss ($\mathcal{L}_{\text{CL}}$) adapts the earlier InfoNCE loss (Oord et al., 2018) and is formulated as:

$$\mathcal{L}_{\text{CL}} = \mathop{\mathbb{E}}_{\substack{p(x,x^+) \\ \{p_n(x_i^-)\}_{i=1}^{|\mathcal{N}_x|}}} \left[ -\log \frac{\exp\left(z \cdot z^+ / \tau\right)}{\sum\limits_{i \in [|\mathcal{N}_x|]} \exp\left(z \cdot z_i^- / \tau\right)} \right] \tag{1}$$

where $\tau > 0$ is the temperature parameter. The minimization of the CL loss contributes to learning an embedding space where samples from the positive pair are pulled closer and samples of the negative pair are pushed apart. However, the CL loss is typically used in the **unsupervised pre-training** (Chen et al., 2020; He et al., 2020; Grill et al., 2020) setting. To adapt it to **domain generalization** (Yao et al., 2022; Chen et al., 2022; Kim et al., 2021), the full model is also required to learn from supervised signals. Thus, it is intuitive to combine the CL loss with the empirical risk minimization (ERM) loss $\mathcal{L}_{\text{ERM}}$ as the following objective:

$$\mathcal{L} = \mathcal{L}_{\text{ERM}} + \lambda \mathcal{L}_{\text{CL}} \tag{2}$$

where $\lambda$ is the regularization parameter. In practice, $\mathcal{L}_{\text{ERM}}$ is usually chosen as the softmax cross entropy loss to classify the output embedding $z$; we follow the classical setting (as well as the previous studies) in this paper. We note that contrastive learning is only performed during training to regularize the learned representations.

## 3 PROPOSED METHODOLOGY

We will shortly revisit the recent theoretical understanding of CL (Wang et al., 2022b), and show how the implications from CL theory motivate the design of `DCCL` for domain generalization.

### 3.1 IMPLICATIONS FROM CONTRASTIVE LEARNING THEORY

We take a recent study on contrastive learning (Wang et al., 2022b) as the main tool to analyze the failure of self-contrastive learning in the previous subsection. Their analysis shows the ERM loss (the pure classification loss) is mainly impacted by the intra-class conditional variance of the learned representation, and the usage of CL can help reduce the intra-class conditional variance, thus controlling the ERM loss.

The magic comes from the intra-class data connectivity enforced by CL. In applying CL, proper data augmentation can help "connect" two different samples $x_i, x_j$ within the same class, which technically means there exists a pair of augmentation maps $a_i, a_j$ so that $a_i(x_i), a_j(x_j)$ are close to each other. As pushed in optimizing the CL loss (1), the ultimate representations $f_h(x_i), f_h(x_j)$ will finally be close since

$$f_h(x_i) \approx f_h(a_i(x_i)) \approx f_h(a_j(x_j)) \approx f_h(x_j).$$

In other words, as a ladder, $a_i(x_i), a_j(x_j)$ connect the two samples $x_i, x_j$, and analogously all the samples within the same class will be connected by proper data augmentation. CL later on pushes their new representations to cluster thanks to the CL loss.

To illustrate the statement above, we construct a toy classification task in Appendix C, where data augmentation is removed. SCL in this example fails to obtain intra-class connectivity due to insufficient data augmentation and domain-separated (rather than class-separated) representations, which ultimately causes poor classification performance. We further remark a similar idea of leveraging the sample similarities in the same class has been studied by Arjovsky et al. (2019, invariant risk minimization), while the CL theory removes the limitation that the marginal distribution on source domains should be the same on target domains, and thus is theoretically more applicable to DG.

### 3.2 MORE AGGRESSIVE DATA AUGMENTATION AND CROSS-DOMAIN POSITIVE SAMPLES

Inspired by the theoretical analysis above, in this subsection we propose two direct approaches to improve intra-class connectivity: (i) applying more aggressive data augmentation and (ii) expanding the scope of positive samples, from solely self-augmented outputs $a(x)$ to the augmentation of intra-class samples across domains.

For the first approach, in spite of the fact that data augmentation in DG (such as horizontal flipping and color jittering) has already been a standard regularization technique (Gulrajani & Lopez-Paz, 2020; Cha et al., 2021; Wang et al., 2022a), the choice of data augmentation, we emphasize, matters for contrastive learning in the domain generalization setting. We naturally need a larger augmentation distribution $\mathcal{A}$ to connect $a_i(x_i)$ and $a_j(x_j)$ since $x_i, x_j$ can be drawn from different domains. As ablation studies, the effect of data augmentation intensity is evaluated in Section 4.3.

Motivated by supervised CL (Khosla et al., 2020; Gunel et al., 2020; Cui et al., 2021), we further introduce **cross-domain** positive pairs into contrastive learning to bridge the intra-class samples scattered in different domains. Specifically, we not only consider the correlated views of the same data sample as positive pairs but also the augmented instances from other intra-class samples across domains. The positive sample $x^+$ will now be conditionally independent of $x$, and the positive pairs have the same conditional distribution $p^{(1)}(x^+|y) = p(x|y)$ [2] (the specific distribution of the positive sample $x^+$ in this subsection will be denoted with a superscript (1)); in other words, $x^+$ can now be the augmentation view of a random sample within the same class $y$ of $x$. With the joint distribution of $x, x^+$ denoted as $p^{(1)}(x, x^+) = \int_y p^{(1)}(x^+|y)p(x|y)p(y)\mathrm{d}y$, the primal domain-

---

[2]Unlike the classical setting in self-supervised CL, in DG we can access the label $y$ in training.

connecting contrastive learning ($\texttt{DCCL}$) objective $\mathcal{L}_{\texttt{DCCL}}^{(0)}$ can be formulated as:

$$\mathcal{L}_{\texttt{DCCL}}^{(0)} = \mathop{\mathbb{E}}_{\substack{p^{(1)}(x,x^+) \\ \{p_n(x_i^-)\}_{i=1}^{|\mathcal{N}_x|}}} \left[ -\log \frac{\exp\left(z \cdot z^+/\tau\right)}{\sum\limits_{i\in[|\mathcal{N}_x|]} \exp\left(z \cdot z_i^-/\tau\right)} \right]. \tag{3}$$

Without the explicit use of domain information, $-\log\exp\left(z \cdot z^+/\tau\right)$, the term corresponding to alignment in loss (3), can now push the intra-class samples from different domains together.

### 3.3 ANCHORING LEARNED MAPS TO PRE-TRAINED MODELS

Up to now, we have not addressed the core difficulty in domain generalization—lack of access to the testing domains in training: CL is originally designed for the self-supervised scenario where a huge amount and wide range of data are fed to the models. However, in the context of domain generalization, the model is just fine-tuned on limited data within partial domains. Consequently, the mechanism of CL can only contribute to the clustering of representations in the seen domains, while the embeddings of the unseen testing domains and the ones of the training domains in the same class may still be separated.

Interestingly, the intra-class connectivity for representations, the desired property in CL, seems to exist at the beginning of the fine-tuning. We observe the phenomenon when visualizing the representations obtained from the pre-trained model using t-SNE (Van der Maaten & Hinton, 2008) in Figure 4a, which thereby motivates our design in this subsection. We can find that mapped by the initial pre-trained model ResNet-50, intra-class samples of the training domains and the testing domains are scattered while well-connected.

We attribute the phenomenon to the effective representations returned by pre-trained model, which reasonably model the pairwise interactions among images and thus draw target domains closer to source domains. To verify the effectiveness of the representations, we design a quantitative **metric to evaluate** whether the pre-trained space is "well-connected", by turning to the concept of "connectivity" in graphs. Details can be found in Appendix A.4.

As for the model design, the phenomenon motivates us to better utilize the pre-trained model $f_{\text{pre}}$ for stronger intra-class connectivity in the mapped representations obtained from $f$. We propose to take the usage of pre-trained models as data augmentation in a disguised form: regular data augmentation works on the raw data and return $x$ while we can further "augment" the representation $x$ via $f_{\text{pre}}$.

In mathematical language, we descibe our design as follows. Upon the augmented sample $x$ defined in the last subsection, we further incorporate the pre-trained embedding $z_{\text{pre}} = h \circ f_{\text{pre}}(x)$ into the definition of feasible positive embeddings $z^{(2),+}$, which expands the scope of the previous positive embeddings $z^+$ (the superscript (2) implies the different distribution compared to $z^+$ in the last subsection). In particular, for a given $x$, we decide the form of the newly coined positive embedding $z^{(2),+}$ as:

$$z^{(2),+} = \begin{cases} z^+ = h \circ f(x^+), & \text{w.p. } \frac{1}{2}, \\ z_{\text{pre}} = h \circ f_{\text{pre}}(x), & \text{w.p. } \frac{1}{2}. \end{cases}$$

With the distribution of the extended positive embedding denoted as $p^{(2)}\left(z^{(2),+}\right)$ (the positive pairs $x, x^+$ still follow $p^{(1)}(x, x^+)$), the proposed $\texttt{DCCL}$ loss $\mathcal{L}_{\texttt{DCCL}}$ can be written as:

$$\mathcal{L}_{\texttt{DCCL}} = \mathop{\mathbb{E}}_{\substack{p^{(2)}\left(z, z^{(2),+}\right) \\ \{p_n(x_i^-)\}_{i=1}^{|\mathcal{N}_x|}}} \left[ -\log \frac{\exp\left(z \cdot z^{(2),+}/\tau\right)}{\sum\limits_{i\in[|\mathcal{N}_x|]} \exp\left(z \cdot z_i^-/\tau\right)} \right], \tag{4}$$

where $p^{(2)}\left(z, z^{(2),+}\right)$ is the joint distribution of $z, z^{(2),+}$ constructed in this subsection.

### 3.4 GENERATIVE TRANSFORMATION LOSS FOR PRE-TRAINED REPRESENTATION

In the previous section, our proposed contrastive learning method manages to mine the supervised signal at the **inter-sample** level, where we align the positive pairs (composed of different samples) while pushing apart the samples in a negative pool.

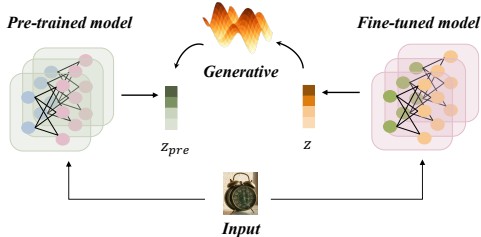

**Figure 3:** An overview of the generative transformation module in DCCL. Two representations $z_{pre}$ and $z$ of the same image are generated via the pre-trained and the fine-tuned model respectively. The variational reconstruction is conducted to encode essential within-sample information.

Echoing the findings in (Yao et al., 2022), which point out that directly aligning positive pairs across vastly different domains often results in poor performance, our research similarly identifies a substantial gap in the representations of pre-trained and fine-tuned models. Direct alignment using contrastive learning as evidenced by our empirical evaluation, tends to be sub-optimal. In response, we introduce the concept of variational generative loss to comprehend the transformation process and bridge these representational gaps. Additionally, the generative transformation module is designed to reconstruct the features of the pre-trained model at an intra-sample level. This complements the inter-sample level supervision provided by contrastive loss. The module, along with its associated loss function, is intended to provide a more enriched supervised signal, encapsulating crucial within-sample information. This module, in turn, supports as a pivotal proxy objective that facilitates model anchoring 3.3.

To simplify the notation of the transformation, we abuse the previous notations $\{z, z_{\text{pre}}\}$ for the output embedding from a certain learned/pre-trained model layer, omitting the corresponding layer denotation. $z_{\text{pre}}$ is the fixed supervised signal provided by the pre-trained model.

With the notation $\{z, z_{\text{pre}}\}$, we introduce the following variational generative model to parameterize the map $g : z \mapsto z_{\text{pre}}$ relating the representation manifolds formed by (the first several layers of) the learned map $f$ and the fixed pre-trained model $f_{\text{pre}}$. In particular, $g$ is composed of an encoder $\phi$ modeling a tunable conditional distribution $q_\phi(z_{\text{lat}} \mid z)$ of $z_{\text{lat}}$ and a tunable decoder $\psi$ mapping $z_{\text{lat}}$ back to $z_{\text{pre}}$, in which $z_{\text{lat}} \in \mathbb{R}^{d'}$ is the latent representation of the generator. Similar to the training of a regular variational autoencoder (VAE) (Kingma et al., 2019), the latent variable $z_{\text{lat}}$ will be sampled from $q_\phi(z_{\text{lat}} \mid z)$; we can then project $z_{\text{lat}}$ to the pre-trained embedding space via decoder $\psi$. Our variational reconstruction loss $\mathcal{L}_{\text{DCCL}}^{\text{Gen}}$ is designed as:

$$\mathcal{L}_{\text{DCCL}}^{\text{Gen}} = -\mathbb{E}_{q_\phi(z_{\text{lat}}|z)}\left[\log p_\psi\left(z_{\text{pre}} \mid z_{\text{lat}}\right)\right] + \text{KL}\left[q_\phi\left(z_{\text{lat}} \mid z\right) \| p\left(z_{\text{lat}}\right)\right], \quad (5)$$

where $p(z_{\text{lat}})$ is the pre-specified prior distribution of $z_{\text{lat}}$, $p_\psi(z_{\text{pre}} \mid z_{\text{lat}})$ is decided by the "reconstruction loss" $\|z_{\text{pre}} - \psi(z_{\text{lat}})\|^2$, and the KL divergence term corresponds to the variational regularization term to avoid mode collapse. The workflow of our proposed generative transformation is shown in Figure 3.

Finally, to benefit the representation learning through both generative transformation and our improved contrastive leaning, we set our ultimate objective as:

$$\mathcal{L} = \mathcal{L}_{\text{ERM}} + \lambda \mathcal{L}_{\text{DCCL}} + \beta \mathcal{L}_{\text{DCCL}}^{\text{Gen}} \quad (6)$$

where $\lambda$ and $\beta$ are coefficients to balance the multi-task loss. The ablation studies in Section 4.3 verify the effectiveness of each component.

## 4 EXPERIMENTS

In this section, we empirically evaluate the performance of our proposed DCCL, intending to answer the following research questions:

- **RQ1:** Does DCCL enable networks to learn transferable representation under distribution shifts?
- **RQ2:** How do different components in our framework contribute to the performance?
- **RQ3:** How good is the generalizability of our proposed DCCL under different circumstances (e.g., varying label ratios and backbones)?
- **RQ4:** Does DCCL really connect the cross-domain representations?

### 4.1 EXPERIMENTAL SETTINGS

We exhaustively evaluate out-of-domain (OOD) accuracy of DCCL on various representative DG benchmarks as in Cha et al. (2021); Yao et al. (2022); Cha et al. (2022); Chen et al. (2022): Office-

| Algorithm | A | C | P | S | Avg. |
|---|---|---|---|---|---|
| IRM (Arjovsky et al., 2019) | 84.8 | 76.4 | 96.7 | 76.1 | 83.5 |
| MetaReg (Balaji et al., 2018) | 87.2 | 79.2 | 97.6 | 70.3 | 83.6 |
| DANN (Ganin et al., 2016) | 86.4 | 77.4 | 97.3 | 73.5 | 83.7 |
| ERM (Vapnik, 1999) | 85.7 | 77.1 | 97.4 | 76.6 | 84.2 |
| GroupDRO (Ganin et al., 2016) | 83.5 | 79.1 | 96.7 | 78.3 | 84.4 |
| MTL (Blanchard et al., 2021) | 87.5 | 77.1 | 96.4 | 77.3 | 84.6 |
| I-Mixup (Xu et al., 2020) | 86.1 | 78.9 | 97.6 | 75.8 | 84.6 |
| MMD (Li et al., 2018b) | 86.1 | 79.4 | 96.6 | 76.5 | 84.7 |
| VREx (Krueger et al., 2021) | 86.0 | 79.1 | 96.9 | 77.7 | 84.9 |
| MLDG (Li et al., 2018a) | 85.5 | 80.1 | 97.4 | 76.6 | 84.9 |
| ARM (Zhang et al., 2020) | 86.8 | 76.8 | 97.4 | 79.3 | 85.1 |
| RSC (Huang et al., 2020) | 85.4 | 79.7 | 97.6 | 78.2 | 85.2 |
| Mixstyle (Zhou et al., 2021) | 86.8 | 79.0 | 96.6 | 78.5 | 85.2 |
| ER (Zhao et al., 2020) | 87.5 | 79.3 | **98.3** | 76.3 | 85.3 |
| pAdaIN (Nuriel et al., 2021) | 85.8 | 81.1 | 97.2 | 77.4 | 85.4 |
| SelfReg (Kim et al., 2021) | 85.6 | 81.0 | 95.9 | 80.5 | 85.6 |
| EISNet (Wang et al., 2020) | 86.6 | 81.5 | 97.1 | 78.1 | 85.8 |
| CORAL (Sun & Saenko, 2016) | 88.3 | 80.0 | 97.5 | 78.8 | 86.2 |
| SagNet (Nam et al., 2021) | 87.4 | 80.7 | 97.1 | 80.0 | 86.3 |
| DSON (Seo et al., 2020) | 87.0 | 80.6 | 96.0 | 82.9 | 86.6 |
| COMEN (Chen et al., 2022) | 88.1 | 82.6 | 97.2 | 81.9 | 87.5 |
| SWAD (Cha et al., 2021) | 89.3 | 83.4 | 97.3 | 82.5 | 88.1 |
| MIRO (Cha et al., 2022) | 89.8 | 83.6 | 98.2 | 82.1 | 88.4 |
| PCL (Yao et al., 2022) | 90.2 | 83.9 | 98.1 | 82.6 | 88.7 |
| Ours | **90.5** | **84.2** | 98.0 | **83.3** | **89.1± 0.1** |

(a) PACS

| Algorithm | A | C | P | R | Avg |
|---|---|---|---|---|---|
| Mixstyle (Zhou et al., 2021) | 51.1 | 53.2 | 68.2 | 69.2 | 60.4 |
| IRM (Arjovsky et al., 2019) | 58.9 | 52.2 | 72.1 | 74.0 | 64.3 |
| ARM (Zhang et al., 2020) | 58.9 | 51.0 | 74.1 | 75.2 | 64.8 |
| RSC (Huang et al., 2020) | 60.7 | 51.4 | 74.8 | 75.1 | 65.5 |
| CDANN (Li et al., 2018b) | 61.5 | 50.4 | 74.4 | 76.6 | 65.7 |
| DANN (Ganin et al., 2016) | 59.9 | 53.0 | 73.6 | 76.9 | 65.9 |
| GroupDRO (Ganin et al., 2016) | 60.4 | 52.7 | 75.0 | 76.0 | 66.0 |
| MMD (Li et al., 2018b) | 60.4 | 53.3 | 74.3 | 77.4 | 66.4 |
| MTL (Blanchard et al., 2021) | 61.5 | 52.4 | 74.9 | 76.8 | 66.4 |
| VREx (Krueger et al., 2021) | 60.7 | 53.0 | 75.3 | 76.6 | 66.4 |
| MLDG (Li et al., 2018a) | 61.5 | 53.2 | 75.0 | 77.5 | 66.8 |
| ERM (Vapnik, 1999) | 63.1 | 51.9 | 77.2 | 78.1 | 67.6 |
| SelfReg (Kim et al., 2021) | 63.6 | 53.1 | 76.9 | 78.1 | 67.9 |
| I-Mixup (Xu et al., 2020) | 62.4 | 54.8 | 76.9 | 78.3 | 68.1 |
| SagNet (Nam et al., 2021) | 63.4 | 54.8 | 75.8 | 78.3 | 68.1 |
| CORAL (Sun & Saenko, 2016) | 65.3 | 54.4 | 76.5 | 78.4 | 68.7 |
| COMEN (Chen et al., 2022) | 65.4 | 55.6 | 75.8 | 78.9 | 68.9 |
| SWAD (Cha et al., 2021) | 66.1 | 57.7 | 78.4 | 80.2 | 70.6 |
| PCL (Yao et al., 2022) | 67.3 | **59.9** | 78.7 | 80.7 | 71.6 |
| MIRO (Cha et al., 2022) | 68.8 | 58.1 | 79.9 | 82.6 | 72.4 |
| Ours | **70.1** | 59.1 | **81.4** | **83.4** | **73.5 ± 0.2** |

(b) Office-Home

Table 1: Experimental comparisons with state-of-the-art methods on benchmarks with ResNet-50. (The tables are re-scaled due to space limit.)

Home (Venkateswara et al., 2017), PACS (Li et al., 2017), VLCS (Fang et al., 2013), TerraIncognita (Beery et al., 2018), and DomainNet (Peng et al., 2019). The details of the data sets are shown in Appendix A.1. For fair comparison, we strictly follow the experimental settings in Gulrajani & Lopez-Paz (2020); Cha et al. (2021); Yao et al. (2022); Chen et al. (2022) and adopt the widely used leave-one-domain-out evaluation protocol, i.e., one domain is chosen as the held-out testing domain and the rest are regarded as source training domains. The experiment results are all averaged over three repeated runs. Following DomainBed (Gulrajani & Lopez-Paz, 2020), we leave 20% of source domain data for validation and model selection. As in previous works (Cha et al., 2022; Yao et al., 2022), we use the ResNet-50 model pre-trained on ImageNet by default, and our code is mainly built upon DomainBed (Gulrajani & Lopez-Paz, 2020) and SWAD (Cha et al., 2021). Due to space constraints, detailed implementation and experimental setups are shown in Appendix A.1. The limitations, attribution of existing assets, and the use of personal data are discussed in Appendix D.

## 4.2 RESULTS (RQ1)

We provide comprehensive comparisons with a set of strong baselines on the domain generalization benchmarks, PACS and OfficeHome, in Tables 1a and 1b. Detailed experimental results on TerraIncognita, VLCS, and DomainNet datasets are deferred to Appendix A.2. We observe our proposed method achieves the best performance: the metrics are 44.0 (ERM)→47.0 (Best Baseline)→47.5 (Ours) on DomainNet, 77.3→79.6→80.0 on VLCS, and 47.8→52.9→53.7 on TerraIncognita. The results of the intermediate columns in the tables represent performance on the testing domain. For example, "A" in Table 1 denotes testing on domain Art and training on Photo, Cartoon, and Sketch. The final result is averaged over all domains. The symbol + in the tables is used to denote that the reproduced experimental performance is clearly distinct from the reported one (such as "PCL+" in Table 4). All the baselines are sorted in ascending order of their performance.

We have the following findings from the tables. (i) We find that DCCL substantially outperforms all the baseline methods concerning OOD accuracy. This indicates the capability of DCCL to extract transferable representation for generalization under distribution shift. (ii) We notice most baselines make explicit use of domain supervision, while only a few methods such as RSC (Huang et al., 2020), SagNet (Nam et al., 2021), COMEN (Chen et al., 2022), SWAD (Cha et al., 2021), MIRO (Cha et al., 2022) and our DCCL do not. The excellent performance of our DCCL may reveal previous works do not well utilize the domain information and there is still much room for improvement. (iii) We note that PCL (Yao et al., 2022) (Proxy Contrastive Learning) has utilized the potential of CL, aligns embeddings of different samples into domain centers, and consistently achieves good performance. Meanwhile, MIRO (Cha et al., 2022) also preserves the pre-trained features by adding the mutual information regularization term and attains satisfactory performance. However, because

| CDC | PMA | GT | A | C | P | R | Avg. |
|---|---|---|---|---|---|---|---|
| - | - | - | 66.1 | 57.7 | 78.4 | 80.2 | 70.6 |
| with Self-Contrast | | | 65.4 | 51.4 | 79.1 | 79.5 | 68.9 |
| ✓ | - | - | 68.0 | 57.9 | 80.1 | 81.3 | 71.8 |
| - | ✓ | - | 68.8 | 57.8 | 80.4 | 82.3 | 72.3 |
| - | - | ✓ | 69.0 | 56.9 | 80.6 | 81.6 | 72.0 |
| - | ✓ | ✓ | 70.0 | 58.7 | 80.5 | 83.4 | 73.1 |
| ✓ | ✓ | - | 69.2 | 58.5 | 81.0 | 83.0 | 72.9 |
| ✓ | - | ✓ | 69.0 | 58.5 | 80.7 | 82.1 | 72.6 |
| w/o Aggressive Aug | | | 69.8 | 58.6 | 81.0 | 82.6 | 73.0 |
| ✓ | ✓ | ✓ | **70.1** | **59.1** | **81.4** | **83.4** | **73.5** |

Table 2: Ablation Studies of `DCCL` on OfficeHome.

| Ratio | Algorithm | A | C | P | R | Avg. |
|---|---|---|---|---|---|---|
| | ERM (Vapnik, 1999) | 40.4 | 32.6 | 42.6 | 49.2 | 41.2 |
| | SWAD (Cha et al., 2021) | 46.9 | 36.2 | 48.5 | 54.2 | 46.4 |
| 5% | COMEN (Chen et al., 2022) | 47.7 | 39.2 | 50.6 | 56.1 | 48.4 |
| | PCL (Yao et al., 2022) | 48.4 | 42.3 | 55.2 | 57.2 | 50.8 |
| | MIRO (Cha et al., 2022) | 51.0 | 41.6 | 58.6 | 61.5 | 53.2 |
| | Ours | **55.7** | **44.1** | **63.1** | **67.1** | **57.5 (+16.3)** |
| | ERM (Vapnik, 1999) | 45.1 | 41.9 | 55.9 | 58.0 | 50.2 |
| | COMEN (Chen et al., 2022) | 50.4 | 44.3 | 56.8 | 60.9 | 53.1 |
| 10% | SWAD (Cha et al., 2021) | 53.3 | 43.9 | 61.8 | 65.2 | 56.1 |
| | PCL (Yao et al., 2022) | 54.6 | 45.1 | 60.9 | 67.2 | 57.0 |
| | MIRO (Cha et al., 2022) | 58.9 | 46.6 | 68.6 | 71.7 | 61.4 |
| | Ours | **62.5** | **49.2** | **72.3** | **75.1** | **64.8 (+14.6)** |

Table 3: Experimental comparisons of `DCCL` with representative baselines on OfficeHome under various label ratios.

of their deficiency to connect cross-domain representations, our method manages to improve upon the success the previous baselines had.

## 4.3 ABLATION STUDIES (RQ2)

In this part, we investigate the effectiveness of the proposed `DCCL` by evaluating the impact of different components. We denote the Cross-Domain Contrastive learning in Section 3.2 as CDC (with more aggressive data augmentation and cross-domain positive samples), Pre-trained Model Anchoring in Section 3.3 as PMA, and Generative Transformation in Section 3.4 as GT. The ablation results are summarized in Table 2. The check mark in the table indicates the module is incorporated. We note that our improved contrastive learning loss in Eqn. (4) has two components: CDC and PMA. The overall improvement of the loss is substantial: $70.6 \rightarrow 72.9$. From the table, we can observe that all the components are useful: when any one of these components is removed, the performance drops accordingly. For example, removing PMA module leads to significant performance degeneration, which verifies the importance of anchoring learned maps to pre-trained models. We can then find the combination of PMA and GT leads to the highest improvement in the ablation, which indicates GT and PMA modules complement each other in an effective way. The finding is also consistent with our motivation in Section 3.4. Moreover, we also evaluate self-contrastive learning. The experimental results indicate that self-contrastive learning will distort the learned embeddings and hamper performance. Besides, the experiment without aggressive data augmentation also validates the effectiveness of stronger data augmentations we suggest in Section 3.2. Additional experimental details and explanations regarding our choices for VAE structures, contrastive learning techniques within `DCCL`, cross-domain examples in CDC, alternative pre-trained backbones, and the Wilds Benchmark can be found in Appendix A.5.

## 4.4 CASE STUDIES

**Generalization ability (RQ3)**. To verify the generalizability of our proposed `DCCL`, we conduct experiments[3] with different label ratios (the percentage of labeled training data) and backbones. (i) In Table 3, we find `DCCL` can obtain consistent improvement over baselines, in both cases of 5% and 10% label ratios. Our method yields a 16.3 and 14.6 absolute improvement compared with

---

[3]We select a few of the most representative methods as baselines.

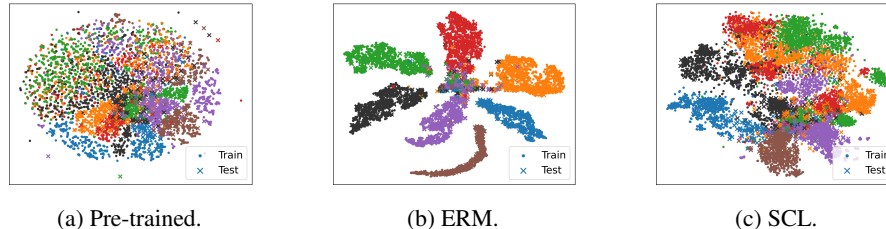

| (a) Pre-trained. | (b) ERM. | (c) SCL. |

Figure 4: t-SNE visualization of the representations across both training and testing domains, output by Pre-trained, ERM, and SCL respectively. Same-class points are in the same colors, and two marker types differentiate the training or the testing domains. We visualize the embedding on PACS dataset where the source domains are Photo, Sketch, and Cartoon; the target domain is Art.

| Algorithm | A | C | P | R | Avg. |
|---|---|---|---|---|---|
| ERM (Vapnik, 1999) | 50.6 | 49.0 | 69.9 | 71.4 | 60.2 |
| SWAD (Cha et al., 2021) | 54.6 | 50.0 | 71.1 | 72.8 | 62.1 |
| PCL$^+$ (Yao et al., 2022) | 58.8 | 51.9 | 74.2 | 75.2 | 65.0 |
| MIRO (Cha et al., 2022) | 59.7 | 52.6 | 75.0 | 77.7 | 66.2 |
| COMEN (Chen et al., 2022) | 57.6 | **55.8** | 75.5 | 76.9 | 66.5 |
| "Mismatch" | 53.4 | 50.7 | 72.3 | 74.0 | 62.6 |
| Ours | **61.7** | 53.6 | **75.9** | **78.7** | **67.5** |

Table 4: Experimental comparisons of `DCCL` on OfficeHome with the ResNet-18 backbone in use.

ERM. We can observe that as the number of available labels reduces, the model benefits more from our `DCCL` (compared with previous 67.6→73.5 increase under 100% label ratio in Table 1b). (ii) In Table 4, we test the performance with a new backbone, ResNet-18 (previously ResNet-50)[4]. We find that even though the baselines' relative ordering changes significantly, our model still performs the best, showcasing the robustness thereof. We further observe replacing the ResNet-18 pre-trained representations to the larger ResNet-50 ones ("mismatch" between the backbone used for fine-tuning and the pre-trained representations) will cause substantial performance drop $67.5 \rightarrow 62.6$.

**Analysis of the representations in `DCCL` (RQ4)**. We analyze the representations returned by `DCCL` to provide more insights. In Figure 4, we utilize t-SNE (Van der Maaten & Hinton, 2008) to visualize the embeddings of the pre-trained model, ERM, and SCL model. We can observe that mapped by the original pre-trained model ResNet-50, the intra-class samples of the training domains and the testing domains are scattered while well-connected. However, in the ERM model, many samples in the testing domain are distributed in the central part of the plot, which is separated from the training samples. There is a clear gap between the training and the testing domains. As for SCL, it seems to harm the learned embedding space and distort the class decision boundary. The observations verify our conjectures in Section C.

We then visualize the embeddings of ERM, PCL, and our `DCCL` methods on the testing domains in Appendix A.3. Our `DCCL` learns discriminative representations even in the unseen target domain by enhancing intra-class connectivity in CL, which is not addressed in ERM and PCL.

## 5 CONCLUSIONS

In this paper, we revisit the role of contrastive learning in domain generalization and identify a key factor: intra-class connectivity. We analyze the failure of directly applying contrastive learning to DG and propose two strategies to improve intra-class connectivity: (i) applying more aggressive data augmentation and (ii) expanding the scope of positive samples. Moreover, to alleviate lack of access to the testing domains in training, we propose to anchor learned maps to pre-trained models which possess the desired connectivity of training and testing domains. Generative transformation is further introduced to complement the pre-trained alignment. Consequently, we combine the pieces together and propose `DCCL` to enable robust representations in the out-of-domain scenario. Extensive experiments on 5 real-world datasets demonstrate the effectiveness of `DCCL`, which outperforms a bundle of baselines.

---

[4]For semantic information matching, pre-trained representations in `DCCL` are generated from the same backbone model used for fine-tuning.

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

## A    DETAILS OF EXPERIMENTS

### A.1    EXPERIMENTAL SETUP

| Datasets | # images | # domains | # classes |
|---|---|---|---|
| PACS | 9991 | 4 | 7 |
| VLCS | 10729 | 4 | 5 |
| OfficeHome | 15588 | 4 | 65 |
| TerraIncognita | 24788 | 4 | 10 |
| DomainNet | 586575 | 6 | 345 |

Table 5: Statistics of datasets.

Here we elaborate the detailed experimental setup of our paper. Following DomainBed (Gulrajani & Lopez-Paz, 2020), we split 80%/20% data from source domains as the training/validation set. The best-performing model on the validation set will be evaluated on the testing target domain to obtain the test performance. The statistics of the experimental datasets are shown in Table 5. We list the number of images, domains, classes in each dataset. The proposed model is optimized using Adam (Kingma & Ba, 2015) with the learning rate of 5e-5. The hyper-parameter $\lambda$ is searched over $\{0.1, 1, 2, 5\}$, and $\beta$ is tuned in the range of $\{0.01, 0.05, 0.1\}$. The temperature $\tau$ is set to 0.1 by default. For the projection head used for contrastive learning, we use a two-layer MLP with ReLU and BatchNorm. Regarding variational reconstruction, following Cha et al. (2022), we employ a simple yet effective architecture, in which the identity function is used as mean encoder and a bias-only network with softplus activation for the variance encoder. More intricate architecture can be explored in the future. Following Gulrajani & Lopez-Paz (2020), for all the datasets except DomainNet, we train the model for 5000 steps. For the DomainNet dataset, we train the model for 15000 steps. Other algorithm-agnostic hyper-parameters such as the batch size are all set to be the same as in the standard benchmark DomainBed (Gulrajani & Lopez-Paz, 2020). For batch construction, we sample the same number of samples from each training domain as in DomainBed (Gulrajani & Lopez-Paz, 2020). Generative Transformation is done for all 4 layers in ResNet-18/50. The experiments are all conducted on one Tesla V100 32 GB GPU. For the data augmentation strategy, previous works usually adopted random cropping, grayscale, horizontal flipping and random color jittering. In this paper, we simply increase the intensity of random color jittering to achieve more aggressive data augmentation. The experimental results have verified the effectiveness of the strategy. Developing stronger and more adaptive augmentation methods for contrastive learning on DG may further enhance the performance.

### A.2    EXPERIMENTAL RESULTS ON TERRAINCOGNITA, VLCS, AND DOMAINNET DATA SETS

We put the experimental comparisons with state-of-the-art baselines on TerraIncognita, VLCS, and DomainNet data sets respectively in Tables 6, 7, and 8. The symbol + in the tables is used to denote that the reproduced experimental performance is distinct from the originally reported one such as "PCL$^{+}$" in Table 8. We can observe our proposed DCCL still surpasses previous methods, which is consistent with the conclusion in the main text and successfully verify the effectiveness of our proposed method.

### A.3    VISUALIZATION

We demonstrate the embeddings of ERM, PCL, and our DCCL methods on the testing domain in Figure 5. ERM, among the three methods, has the most samples distributed in the central area which cannot be distinguished. For the embedding of contrastive-learning-based baseline PCL, there are fewer samples distributed ambiguously. However, the class clusters are not compact and the class boundaries are not clear. By contrast, our DCCL learns discriminative representations even in the unseen target domain by enhancing intra-class connectivity in CL.

| Algorithm | L100 | L38 | L43 | L46 | Avg. |
|---|---|---|---|---|---|
| MMD (Li et al., 2018b) | 41.9 | 34.8 | 57.0 | 35.2 | 42.2 |
| GroupDRO (Ganin et al., 2016) | 41.2 | 38.6 | 56.7 | 36.4 | 43.2 |
| Mixstyle (Zhou et al., 2021) | 54.3 | 34.1 | 55.9 | 31.7 | 44.0 |
| ARM (Zhang et al., 2020) | 49.3 | 38.3 | 55.8 | 38.7 | 45.5 |
| MTL (Blanchard et al., 2021) | 49.3 | 39.6 | 55.6 | 37.8 | 45.6 |
| CDANN (Li et al., 2018b) | 47.0 | 41.3 | 54.9 | 39.8 | 45.8 |
| VREx (Krueger et al., 2021) | 48.2 | 41.7 | 56.8 | 38.7 | 46.4 |
| RSC (Huang et al., 2020) | 50.2 | 39.2 | 56.3 | 40.8 | 46.6 |
| DANN (Ganin et al., 2016) | 51.1 | 40.6 | 57.4 | 37.7 | 46.7 |
| SelfReg (Kim et al., 2021) | 48.8 | 41.3 | 57.3 | 40.6 | 47.0 |
| IRM (Arjovsky et al., 2019) | 54.6 | 39.8 | 56.2 | 39.6 | 47.6 |
| CORAL (Sun & Saenko, 2016) | 51.6 | 42.2 | 57.0 | 39.8 | 47.7 |
| MLDG (Li et al., 2018a) | 54.2 | 44.3 | 55.6 | 36.9 | 47.8 |
| ERM (Vapnik, 1999) | 54.3 | 42.5 | 55.6 | 38.8 | 47.8 |
| I-Mixup (Xu et al., 2020) | 59.6 | 42.2 | 55.9 | 33.9 | 47.9 |
| SagNet (Nam et al., 2021) | 53.0 | 43.0 | 57.9 | 40.4 | 48.6 |
| COMEN (Chen et al., 2022) | 56.0 | 44.3 | 58.4 | 39.4 | 49.5 |
| SWAD (Cha et al., 2021) | 55.4 | 44.9 | 59.7 | 39.9 | 50.0 |
| PCL (Yao et al., 2022) | 58.7 | 46.3 | 60.0 | 43.6 | 52.1 |
| MIRO (Cha et al., 2022) | 60.9 | 47.6 | 59.5 | 43.4 | 52.9 |
| Ours | **62.2** | **48.3** | 60.6 | 43.6 | **53.7 ± 0.2** |

Table 6: Experimental comparisons with state-of-the-art methods on TerraIncognita benchmark with ResNet-50.

| Algorithm | C | L | S | V | Avg |
|---|---|---|---|---|---|
| GroupDRO (Ganin et al., 2016) | 97.3 | 63.4 | 69.5 | 76.7 | 76.7 |
| RSC (Huang et al., 2020) | 97.9 | 62.5 | 72.3 | 75.6 | 77.1 |
| MLDG (Li et al., 2018a) | 97.4 | 65.2 | 71.0 | 75.3 | 77.2 |
| MTL (Blanchard et al., 2021) | 97.8 | 64.3 | 71.5 | 75.3 | 77.2 |
| ERM (Vapnik, 1999) | 98.0 | 64.7 | 71.4 | 75.2 | 77.3 |
| I-Mixup (Xu et al., 2020) | 98.3 | 64.8 | 72.1 | 74.3 | 77.4 |
| MMD (Li et al., 2018b) | 97.7 | 64.0 | 72.8 | 75.3 | 77.5 |
| CDANN (Li et al., 2018b) | 97.1 | 65.1 | 70.7 | 77.1 | 77.5 |
| ARM (Zhang et al., 2020) | 98.7 | 63.6 | 71.3 | 76.7 | 77.6 |
| SagNet (Nam et al., 2021) | 97.9 | 64.5 | 71.4 | 77.5 | 77.8 |
| SelfReg (Kim et al., 2021) | 96.7 | 65.2 | 73.1 | 76.2 | 77.8 |
| Mixstyle (Zhou et al., 2021) | 98.6 | 64.5 | 72.6 | 75.7 | 77.9 |
| PCL (Yao et al., 2022) | 99.0 | 63.6 | 73.8 | 75.6 | 78.0 |
| VREx (Krueger et al., 2021) | 98.4 | 64.4 | 74.1 | 76.2 | 78.3 |
| COMEN (Chen et al., 2022) | 98.5 | 64.1 | 74.1 | 77.0 | 78.4 |
| IRM (Arjovsky et al., 2019) | 98.6 | 64.9 | 73.4 | 77.3 | 78.6 |
| DANN (Ganin et al., 2016) | 99.0 | 65.1 | 73.1 | 77.2 | 78.6 |
| CORAL (Sun & Saenko, 2016) | 98.3 | **66.1** | 73.4 | 77.5 | 78.8 |
| SWAD (Cha et al., 2021) | 98.8 | 63.3 | 75.3 | 79.2 | 79.1 |
| MIRO (Cha et al., 2022) | 98.8 | 64.2 | 75.5 | 79.9 | 79.6 |
| Ours | **99.1** | 64.0 | **76.1** | **80.7** | **80.0 ± 0.1** |

Table 7: Experimental comparisons with state-of-the-art methods on VLCS benchmark with ResNet-50.

## A.4 REPRESENTATION CONNECTIVITY OF PRE-TRAINED MODELS

Our motivation to utilize pre-trained models for better connectivity is intuitive: we consider pre-trained model can return effective representations modeling the pairwise interactions among images, which thus draws target domains closer to source domains. To verify the motivation, we conduct experiments to evaluate whether the pre-trained model is "well-connected".

1. We design a quantitative **metric to help evaluate** whether the pre-trained space is "well-connected". For images within the same class, we take those images as nodes and construct a graph, only connecting two nodes when their distance on the pre-trained space is smaller than

| Algorithm | clip | info | paint | quick | real | sketch | Avg |
|---|---|---|---|---|---|---|---|
| MMD (Li et al., 2018b) | 32.1 | 11.0 | 26.8 | 8.7 | 32.7 | 28.9 | 23.4 |
| GroupDRO (Ganin et al., 2016) | 47.2 | 17.5 | 33.8 | 9.3 | 51.6 | 40.1 | 33.3 |
| VREx (Krueger et al., 2021) | 47.3 | 16.0 | 35.8 | 10.9 | 49.6 | 42.0 | 33.6 |
| IRM (Arjovsky et al., 2019) | 48.5 | 15.0 | 38.3 | 10.9 | 48.2 | 42.3 | 33.9 |
| Mixstyle (Zhou et al., 2021) | 51.9 | 13.3 | 37.0 | 12.3 | 46.1 | 43.4 | 34.0 |
| ARM (Zhang et al., 2020) | 49.7 | 16.3 | 40.9 | 9.4 | 53.4 | 43.5 | 35.5 |
| CDANN (Li et al., 2018b) | 54.6 | 17.3 | 43.7 | 12.1 | 56.2 | 45.9 | 38.3 |
| DANN (Ganin et al., 2016) | 53.1 | 18.3 | 44.2 | 11.8 | 55.5 | 46.8 | 38.3 |
| RSC (Huang et al., 2020) | 55.0 | 18.3 | 44.4 | 12.2 | 55.7 | 47.8 | 38.9 |
| I-Mixup (Xu et al., 2020) | 55.7 | 18.5 | 44.3 | 12.5 | 55.8 | 48.2 | 39.2 |
| SagNet (Nam et al., 2021) | 57.7 | 19.0 | 45.3 | 12.7 | 58.1 | 48.8 | 40.3 |
| MTL (Blanchard et al., 2021) | 57.9 | 18.5 | 46.0 | 12.5 | 59.5 | 49.2 | 40.6 |
| MLDG (Li et al., 2018a) | 59.1 | 19.1 | 45.8 | 13.4 | 59.6 | 50.2 | 41.2 |
| CORAL (Sun & Saenko, 2016) | 59.2 | 19.7 | 46.6 | 13.4 | 59.8 | 50.1 | 41.5 |
| SelfReg (Kim et al., 2021) | 60.7 | 21.6 | 49.4 | 12.7 | 60.7 | 51.7 | 42.8 |
| MetaReg (Balaji et al., 2018) | 59.8 | **25.6** | 50.2 | 11.5 | 64.6 | 50.1 | 43.6 |
| DMG (Chattopadhyay et al., 2020) | 65.2 | 22.2 | 50.0 | 15.7 | 59.6 | 49.0 | 43.6 |
| ERM (Vapnik, 1999) | 63.0 | 21.2 | 50.1 | 13.9 | 63.7 | 52.0 | 44.0 |
| COMEN (Chen et al., 2022) | 64.0 | 21.1 | 50.2 | 14.1 | 63.2 | 51.8 | 44.1 |
| PCL$^+$ (Yao et al., 2022) | 64.3 | 20.9 | 52.7 | **16.7** | 62.2 | 55.5 | 45.4 |
| SWAD (Cha et al., 2021) | 66.0 | 22.4 | 53.5 | 16.1 | 65.8 | 55.5 | 46.5 |
| MIRO (Cha et al., 2022) | 66.4 | 23.5 | 54.1 | 16.2 | 66.8 | 54.8 | 47.0 |
| Ours | **66.9** | 23.0 | **55.1** | 16.0 | **67.7** | **56.1** | **47.5 ± 0.0** |

Table 8: Experimental comparisons with state-of-the-art methods on DomainNet benchmark with ResNet-50.

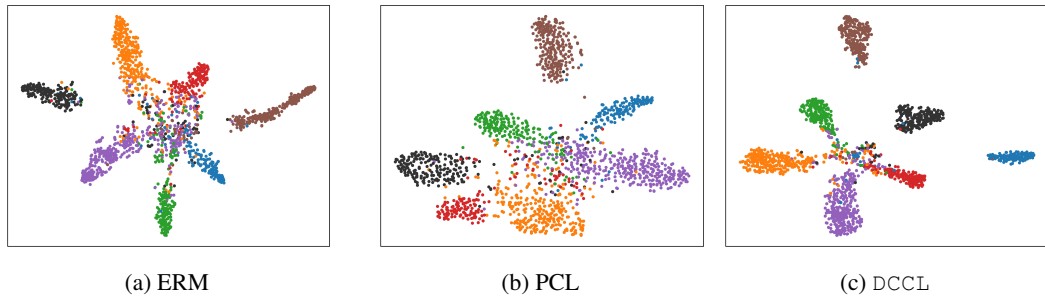

(a) ERM                           (b) PCL                           (c) DCCL

Figure 5: t-SNE visualization of the ERM, PCL and DCCL representations on the testing domain. Same-class points are in the same colors. We visualize the embedding on PACS dataset where the source domains are photo, sketch, and cartoon; the target domain is art.

a threshold. We denote the smallest possible threshold which makes the graph **connected** as $\tau$, and denote the mean and the std of the pairwise distances respectively as $\mu$ and $\sigma$. We can thus use $(\tau - \mu)/\sigma$ as a metric to describe the connectivity of the representations.

2. We report the mean (max) metrics (the smaller, the better) of each class for ERM and pre-trained model on PACS, VLCS, and Terra.; the values for ERM are 1.37 (2.68), 1.78 (2.15), and 3.31 (3.56), for pre-trained model 0.54 (0.81), 0.46 (0.62), and 0.63 (0.76). The results confirm the pre-trained space is well-connected.

Furthermore, the variation in performance improvement across different datasets can be attributed to differences in connectivity. We define a measure to evaluate connectivity in Appendix A.4 where lower values indicate better connectivity. For the pre-trained (ERM) model, the connectivity measure we have is 0.54 (1.37) for PACS and 0.49 (2.85) for OfficeHome. A larger discrepancy in connectivity between ERM and the pretraine model ($\frac{1.37}{0.54}$ v.s. $\frac{2.85}{0.49}$) allows for greater potential for improvement.

## A.5 FURTHER ABLATION STUDY

**Choices of VAE structures**. In our experiments, using more advanced VAE structures like HFVAE (Esmaeili et al., 2019) (72.7) and IntroVAE (Huang et al., 2018) (73.1) will yield worse results than vanilla VAE (73.5), which may be attributed to the increased training difficulty.

**Choices of contrastive learning methods**. SimCLR is denoted as "SelfContrast" in Table 4. Our proposed DCCL (73.5) turns out to outperform other representative SSL approaches: SimCLR (Chen et al., 2020) (68.9 in Tab. 4), MoCo (He et al., 2020) (69.7), BYOL (Grill et al., 2020) (70.7), SwAV (Caron et al., 2020) (71.5).

**Further justification of cross-domain contrast (CDC)**. To further justify cross-domain contrast (CDC), we also implement a baseline using within-domain positive samples only, and the accuracy drops remarkably compared to CDC (71.8 → 70.4). In addition, we include an oracle experiment with solely cross-domain positive pairs and observe comparable performance (71.8 → 71.9). It may require careful design to make good use of domain information to obtain improvements.

**Choices of pre-trained backbone and resources.** In Table 9, we present additional experiments on Instagram (3.6B) pre-trained RegNet. Compared to PCL, which ignores the pre-trained information, DCCL achieves consistent and substantial improvement on imagenet pre-trained models. And when applied to Instagram, the improvement becomes remarkably larger. These indicate the importance of the pre-trained information, and more abundant the pre-training resources, the stronger the pre-trained information is needed.

| Backbone | ResNet-18 | ResNet-50 | RegNet |
|---|---|---|---|
| Resource | ImageNet (1.3M) | | Instagram (3.6B) |
| PCL | 65.0 | 71.6 | 73.2 |
| DCCL | **67.5** (+2.5) | **73.5** (+1.9) | **82.5** (+9.3) |

Table 9: Perf with different pre-trained resources.

**Further Experiments on the Wilds Benchmark.**

We also test the OOD performance of our proposed DCCL using the Camelyon and iWildCam datasets from the Wilds benchmark with the pre-trained ResNet-50 network. In Table 10, DCCL demonstrate a consistent and substantial improvement in performance on the more challenging datasets.

| Datasets | Camelyon | | iWildCam |
|---|---|---|---|
| Metrics | Avg. Acc | Worst Acc | F1 |
| ERM | 88.7 | 68.3 | 31.3 |
| PCL | 91.2 | 75.5 | 30.2 |
| DCCL | **96.7** | **90.9** | **32.7** |

Table 10: Perf on Wilds datasets with pre-trained ResNet-50.

**Further Ablation Study on the VLCS dataset.**

Here we additionally performed an ablation study on the VLCS dataset, as shown in Table 11, where the performance gain above SWAD is relatively smaller. These results further confirm that the three components we identified contribute consistently to the effectiveness, as detailed in our paper.

## B RELATED WORK

In this section, we review the related works in domain generalization and contrastive learning.

### B.1 DOMAIN GENERALIZATION

The goal of DG is to enable models to generalize to unknown target domains under distribution shifts. The related literature can be split into several categories as follows.

| Algorithm | C | L | S | V | Avg |
|---|---|---|---|---|---|
| SWAD | 98.8 | 63.3 | 75.3 | 79.2 | 79.1 |
| DCCL w/o CDC | 98.9 | 63.8 | 75.6 | 79.5 | 79.4 |
| DCCL w/o PMA | 98.6 | 63.7 | 75.7 | 79.3 | 79.3 |
| DCCL w/o GT | 98.7 | **64.3** | 75.2 | 80.2 | 79.6 |
| DCCL | **99.1** | 64.0 | **76.1** | **80.7** | **80.0** |

Table 11: Ablation Study on VLCS dataset with pre-trained ResNet-50.

(i) The first line of work focuses on learning policies. One strategy is meta learning (Finn et al., 2017), which adapts to new environments rapidly with limited observations; the meta-optimization idea was thus introduced in DG (Li et al., 2018a; Balaji et al., 2018; Qiao et al., 2020) to generalize to future testing environments/domains; another widely-studied strategy is ensemble learning (Cha et al., 2021; Chu et al., 2022), claiming DG can benefit from several diverse neural networks to obtain more robust representations. (ii) The second line of work is data augmentation. Many fabricated or learnable augmentation strategies (Volpi et al., 2018; Zhou et al., 2020b; Li et al., 2021; Xu et al., 2020) were developed to regularize and enhance deep learning models. In our paper, we verify more aggressive augmentation can lead to better representations in CL as well. (iii) The last series of work is domain invariant learning. Researchers seek to learn invariances across multiple observed domains for improved generalization on target domains. The commonly used approaches include domain discrepancy regularization (Li et al., 2018b; Zhou et al., 2020a) and domain adversarial learning (Li et al., 2018c; Ganin et al., 2016; Matsuura & Harada, 2020). Recently, MIRO (Cha et al., 2022) began to explore the retention of pre-trained features by designing the mutual information regularization term. The paper (Liu et al., 2023) also utilized the concept connectivity to build up the method. However, their concept of "connectivity" based on joint distribution clearly differ from our paper. Therefore the theoretical motivation behind two papers are indeed different. Moreover, the methods proposed are different. Except for the common strategy of strong augmentation recommended by the contrastive learning theory paper Wang et al. (2022b), our proposed methods are different from the ones in Liu et al. (2023). They propose two nearest-neighbor-based methods for constructing positive pairs, while our main contribution lies in the exploitation of both the pre-trained models and the intra-class data connectivity.

## B.2 CONTRASTIVE LEARNING

Contrastive learning (CL) (Chen et al., 2020) aims to learn discriminative sample representation by aligning positive instances and pushing negative ones apart. As a promising self-supervised learning paradigm, CL is widely used in unsupervised pre-training to improve the performance of downstream tasks (Hjelm et al., 2019; Gao et al., 2021; Li et al., 2022; He et al., 2020; Chen et al., 2020; Caron et al., 2020; Chen & He, 2021; Grill et al., 2020). SimCLR (Chen et al., 2020) is the CL framework that first reveals the projection head and data augmentation as the core components to learn invariant representation across views. MoCo (He et al., 2020) proposes to build a dynamic queue dictionary to enlarge batch size for effective learning. There are also works (Khosla et al., 2020; Gunel et al., 2020; Cui et al., 2021) adapting CL to the supervised setting to leverage label information.

The capability of CL to obtain class-separated representations has also motivated the application in domain generalization. SelfReg (Kim et al., 2021) introduced a new regularization method to build self-supervised signals with only positive samples; PCL (Yao et al., 2022) proposed a proxy-based approach to alleviate the positive alignment issue in CL; COMEN (Chen et al., 2022) used a prototype-based CL component to learn the relationships between various hidden clusters. However, the role of CL in domain generalization is not yet well explored, and our work is dedicated to shedding some light on the understanding of its effect from a intra-class connectivity perspective.

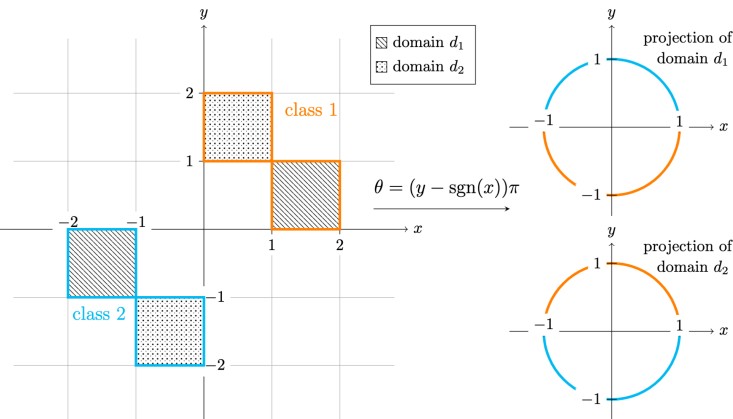

Figure 6: Illustration for the toy example of self-contrastive learning (SCL). Spots and slashes are filled in to represent different domains; orange and blue rectangles respectively denote classes 1 and 2. The mapping function $\varphi \circ \theta$ learned on domain $d_1$ can perfectly classify the samples, and the mapping attains perfect alignment and uniformity (the objective of SCL). However, when applied to a new domain $d_2$, the classifier completely fails (0% acc).

## C    FAILURE OF SELF-CONTRASTIVE LEARNING IN DOMAIN GENERALIZATION

Self-contrastive learning, which aligns the augmentation views of the same input, has achieved successful performance in unsupervised pre-training tasks (Chen et al., 2020; He et al., 2020; Grill et al., 2020). However, it does not naturally fit the domain generalization setting since it assumes the ability to sample $x$ from the whole data distribution; in the training stage of domain generalization, we instead are only able to access partial domains. This mismatch can lead to suboptimal performance in domain generalization if the users mechanically adopt the classical contrastive learning loss.

We provide a linearly separable toy example in Figure 6 to show the deficiency of SCL that even attaining optimal CL loss (1) cannot guarantee good performance in the domain generalization setting, where only partial domains are involved in the training. In the figure, slashes and spots are used to represent domains $d_1$ and $d_2$; orange and blue rectangles respectively denote classes 1 and 2. We specifically consider the extreme case that **no** augmentation is applied and only domain $d_1$ is involved in the training. We then construct a map $\varphi\left(\theta(x)\right) := \left(\cos\left(\theta\right), \sin\left(\theta\right)\right)$ with $\theta(x) = (x - \text{sgn}(y))\pi$ [5]. The map $f_h = \varphi \circ \theta$ attains perfect alignment (due to no augmentation) and maximal uniformity (new representations are uniformly distributed on the corresponding circle arcs) on the 1-sphere $\mathbb{S}^1 := \left\{x \in \mathbb{R}^2 : \|x\|_2 = 1\right\}$, and based on the derivation in Wang & Isola (2020) $f_h$ will minimize the CL loss (1). However, the new representations for domain $d_2$ do not reflect the class information and even have the opposite signs as domain $d_1$.

We can conclude that the usage of classical SCL does not necessarily lead to good performance under the domain generalization setting; and empirical verification is provided in Section 4.4 as well. Similar limitation is observed in invariance-based DG methods (Shui et al., 2022). We provide the detailed settings of the coined data distribution as follows.

**Example C.1** (Self-contrastive learning does not help domain generalization.)**.** *Let the label collection $\mathcal{Y}$ be $\{-1, 1\}$ and the portions of two classes are both $0.5$. Assume there are two domains $d_1$ and $d_2$: if a sample $X = (X_1, X_2) \in \mathbb{R}^2$ with label $Y$ is from domain $d_1$, its conditional distribution will be specified as*

$$\begin{cases} X_1 \sim \text{Unif}\,(0, 1)\, Y, \\ X_2 \sim \text{Unif}\,(1, 2)\, Y, \\ X_1 \perp\!\!\!\perp X_2 \mid Y; \end{cases}$$

*in domain $d_2$ the distribution of $X_1, X_2$ is interchanged. Considering the extreme case that **no** augmentation is applied and only domain $d_1$ is involved in the training, we construct a map*

---

[5] $\text{sgn}(y) := \mathbf{1}_{\{y \geq 0\}} - \mathbf{1}_{\{y < 0\}}$ is the sign function.

$\varphi\left(\theta(x)\right) := \left(\cos\left(\theta\right), \sin\left(\theta\right)\right)$ *with* $\theta(x) = (x_1 - \mathrm{sgn}(y))\,\pi$ [6]. *The map* $f_h = \varphi \circ \theta$ *attains perfect alignment (due to no augmentation) and maximal uniformity (new representations are uniformly distributed on the corresponding circle arcs) on the 1-sphere* $\mathbb{S}^1 := \left\{x \in \mathbb{R}^2 : \|x\|_2 = 1\right\}$, *and based on the derivation in Wang & Isola (2020)* $f_h$ *will minimize the CL loss (1). However, the new representations for domain* $d_2$ *do not reflect the class information and even have the opposite signs as domain* $d_1$.

SCL in the previous example fails to obtain **intra-class connectivity** due to insufficient data augmentation and domain-separated (rather than class-separated) representations, which ultimately causes poor generalization performance. Inspired by the above analysis, we thus propose two approaches to improve intra-class connectivity: (i) applying more aggressive data augmentation and (ii) expanding the scope of positive samples, from solely self-augmented outputs $a(x)$ to the augmentation of intra-class samples across domains.

## D   DISCUSSIONS & LIMITATIONS

In the paper, We analyze the failure of directly applying SCL to DG with the CL theory and suggest lack of intra-class connectivity in the DG setting causes the deficiency. We accordingly propose domain-connecting contrastive learning (DCCL) to enhance the connectivity across domains and obtain generalizable and transferable representation for DG. Extensive experiments also verify the effectiveness of our method.

However, we're also aware of the **limitations** of our work. We don't make explicit use of the domain information. It implies if one can well leverage the domain information, better generalization performance might be obtained. Moreover, similar to Cha et al. (2022), our proposed DCCL requires the pre-trained embeddings of the samples. This existing drawback can be mitigated by generating the pre-trained embeddings in advance and storing them locally. In addition, how to develop stronger and more adaptive augmentation methods for contrastive learning on DG is not explored in this paper and remains an open problem.

Regarding **attribution of existing assets**, we only utilize existing open-sourced datasets, which all can be found in DomainBed[7] benchmark. In addition, we don't make any use of **personal data**. For all the datasets used, there is no private personally identifiable information or offensive content.

---

[6]$\mathrm{sgn}(y) := \mathbf{1}_{\{y \geq 0\}} - \mathbf{1}_{\{y < 0\}}$ is the sign function.
[7]https://github.com/facebookresearch/DomainBed

