# OpenReview forum: "Connecting Domains and Contrasting Samples: A Ladder for Domain Generalization"
_ICLR.cc/2024/Conference — Submitted to ICLR 2024_

### Official Review · Reviewer_2QYf · 2023-10-28

**Soundness:** 2 fair
**Presentation:** 3 good
**Contribution:** 2 fair
**Rating:** 5
**Confidence:** 4

**Summary:**

This paper addresses the challenge of distribution shifts between training and testing datasets in domain generalization (DG). Contrary to expectations, applying contrastive learning (CL) directly in DG settings leads to performance deterioration due to the lack of domain connectivity. To address this, the authors propose domain-connecting contrastive learning (DCCL), which introduces aggressive data augmentation and cross-domain positive samples to improve domain connectivity. They also propose model anchoring to exploit domain connectivity in pre-trained representations. Experimental results on standard DG benchmarks demonstrate that DCCL outperforms state-of-the-art baselines, even without domain supervision.

**Strengths:**

1. The paper presents the paper clearly, making it accessible to a wide range of readers. The exploration of why contrastive learning (CL) is detrimental to Domain Generalization is an intriguing topic.
2. The paper offers a noteworthy contribution by highlighting the finding that pre-trained models with better domain connectivity can lead to improved performance. The authors provide a clear and reasonable definition of domain connectivity, enhancing our understanding of this important aspect.
3. Through extensive experiments on standard Domain Generalization benchmarks, the authors demonstrate the effectiveness of their proposed domain-connecting contrastive learning (DCCL) method, surpassing state-of-the-art baselines.

**Weaknesses:**

1. One potential weakness of the paper is that it may overstate its claims regarding the theoretical analysis of why Contrastive Learning harms the performance of Domain Generalization models. The current version lacks rigorous theoretical analysis and relies more on heuristic motivation rather than providing concrete theoretical explanations.
2. Although the paper proposes improving the similarity of representations across different domains as a key contribution, similar ideas have been previously proposed and utilized in the literature, such as in [1]. This could be seen as a limitation in terms of originality.
3. The proposed Generative Transformation Loss for Pre-trained Representations raises concerns. The approach of fine-tuning the model to align $z$ and $z^{pre}$ appears counterintuitive, as it may reduce performance when using $z$ for prediction. This introduces a potential trade-off between the ERM loss and the Generative Transformation Loss for DCCL, which needs to be carefully considered.
4. The paper fails to adequately explain the relationship between domain connectivity and the classification/regression performance, which is the ultimate goal of Domain Generalization. While domain connectivity is an important metric to consider, the authors do not sufficiently address how it directly impacts the predictive performance of the models in real-world scenarios.

[1] Improving Out-of-Distribution Robustness via Selective Augmentation. ICML'22

**Questions:**

Please see the weakness section.

---

> ### Author Response · Authors · 2023-11-21
> **Response to Reviewer 2QYf**
>
> We appreciate your helpful feedback. We hope the following answers can address your concerns.
>
>
> > **Q1:** It may overstate its claims regarding the theoretical analysis of why Contrastive Learning harms the performance of Domain Generalization models.
>
> We hope to argue that our claim regarding the theoretical analysis is conservative. It's important to note that our paper's central theme is not just introducing the concept of connectivity and theoretical analysis, but rather emphasizing connectivity enforcement **from both data and model perspectives**.
>
> Meanwhile, to clearly convey the main contribution of our paper to the audience, we don't present any decorative theorems/propositions in the paper.
>
> We instead aim to propose useful DG methods for practitioners and attribute most of the theoretical analysis to the motivating theory paper [2]. We agree we leverage the theoretical motivation behind [2], as clearly indicated in Section 3.1, for the new setting of DG; if the readers are interested in the theoretical derivative, they will find the analysis in [2] can be applied to DG settings with proper adaptations and regularity conditions. We do not take that as our contribution, while as suggested by the reviewer we've added a remark to the next revision regarding the theoretical analysis.
>
> > **Q2:** The idea of improving the similarity of representations across different domains is proposed and utilized in the literature.
>
> We want to emphasize the key idea of our paper is not aligning the representations across different domains, but **a unified approach**, combining data and model perspectives with self-supervised objectives to bolster **connectivity**, thereby advancing domain generalization.
>
> From the data angle, we build upon the shortcomings of traditional methods that align augmentation views of the same input, as explained in Appendix C. We propose two direct improvements for enhancing domain connectivity through data:
> 1. implementing more aggressive data augmentation, and
> 2. broadening the range of positive samples, extending beyond self-augmented outputs to include intra-class samples across different domains. Nevertheless, this data alignment strategy merely enhances the clustering of representations within known domains, and may not effectively bridge the gap between unseen test domains and trained domains within the same class.
>
> Therefore, we extend our focus to the model perspective. In the paper, we observed that using a pre-trained ResNet-50 model, intra-class samples from training and testing domains are scattered yet interconnected. By integrating this observation, our paper presents a holistic approach.
>
> > **Q3:** The proposed Generative Transformation Loss for Pre-trained Representations raises concerns.
>
> Thanks for bringing up the issue. We'd like to clarify the role of generative transformation as a pivotal proxy objective that facilitates model anchoring. We've described the details in our [general response](https://openreview.net/forum?id=3mXJ9o2DNx&noteId=X8nuMF5cH6) and incorporated them in Section 3.4.
>
>
>
> > **Q4:** The paper fails to adequately explain the relationship between domain connectivity and classification/regression performance.
>
> We will add the following remark to the next revision for better clarity, while we also hope to argue in the first paragraph of Section 3.1 we have clearly indicated the relationship between domain (intra-class) connectivity and the classification/regression performance is already studied in [2].
>
>
> We remark that after connecting different domains in the DG settings, we can directly follow the derivation in the contrastive learning theory paper [2] and similarly conclude the classification/regression performance benefits from connectivity. Rather than the theoretical analysis, the primary aim of our paper is to thoroughly improve domain connectivity, addressing both data and model aspects, and ultimately bolstering domain generalization capabilities.
>
>
> [1] Promoting Semantic Connectivity: Dual Nearest Neighbors Contrastive Learning for Unsupervised Domain Generalization. CVPR 2023.
>
>
> [2] Chaos is a Ladder: A New Theoretical Understanding of Contrastive Learning via Augmentation Overlap. ICLR 2022.

---

> > ### Comment · Reviewer_2QYf · 2023-11-22
> >
> > Thank you for your response. I have carefully reviewed your feedback as well as the other reviews. I appreciate the authors' rebuttal and understand that emphasizing connectivity enforcement from both data and model perspectives is a novel idea. However, I think that the current version of the paper still requires substantial revisions, such as incorporating the provided clarifications, in order to enhance the clarity of the motivation. Therefore, I have made the decision to maintain my current rating.

---

> > > ### Author Response · Authors · 2023-11-22
> > >
> > > Thank you for your prompt response. We have carefully revised our paper, integrating all our suggestions and clarifications. These revisions are marked in blue for easy reference. We are grateful for your feedback and remain committed to further enhancing the quality of our paper.

---

### Official Review · Reviewer_fbsW · 2023-10-31

**Soundness:** 3 good
**Presentation:** 3 good
**Contribution:** 2 fair
**Rating:** 5
**Confidence:** 4

**Summary:**

This paper focuses on self-supervised learning, especially contrastive learning for domain generalization settings. They analyze the phenomenon with the CL theory and discover the lack of domain connectivity in the DG setting causes the deficiency. Thus they propose domain-connecting contrastive learning (DCCL) to enhance the conceptual connectivity across domains and obtain generalizable representations for
DG.  Some data augmentation strategies are introduced. The experiments demonstrate the effectiveness of the proposed method.

**Strengths:**

1. I think applying contrastive learning to DG is very interesting, and might inspire more work in this direction.
2. The proposed idea is very simple but effective.

**Weaknesses:**

1. My main concern is the discussion of domain connectivity. Is this connectivity based on sample connectivity? Recent work, such as [1], also proposes similar connectivity for domain adaptation but is overlooked in the paper.

2. What is the high-level motivation for the generative transformation loss？Is it necessary to use augmentation of the same sample, acquiring the class knowledge more effectively?


[1] Connect, Not Collapse: Explaining Contrastive Learning for Unsupervised Domain Adaptation, ICML 2022

**Questions:**

Please refer to the above weakness.

---

> ### Author Response · Authors · 2023-11-21
> **Response to Reviewer fbsW**
>
> We appreciate your helpful feedback. We hope the following answers can address your concerns.
>
> > **Q1:** Is this connectivity based on sample connectivity? The discussion with [1] is overlooked.
>
> Our definition of connectivity is based on cross-domain sample connectivity within the class. In [1], the assumption is that data adheres to the **stochastic block model (SBM)**  and they use SBM's edge probability to define connectivity. We've referenced [1] and expanded our discussion in relation to [1] to better articulate our approach.
> It's important to note that our paper's central theme is not just introducing the concept of connectivity, but rather emphasizing its enforcement **from both data and model perspectives**.
>
> For an intuitive understanding of the connectivity in our paper, we introduced another quantitative metric to help evaluate whether the representation space is ''well-connected'' in Appendix A.4. For images within the same class, we take those images as nodes and construct a graph, only connecting two nodes when their distance on the pre-trained space is smaller than a threshold. We denote the smallest possible threshold which makes the graph \textbf{connected} as $\tau$, and denote the mean and the std of the pairwise distances respectively as $\mu$ and $\sigma$. We can thus use $(\tau - \mu)/\sigma$ as a metric to describe the connectivity of the representations.
>
> > **Q2:** What is the high-level motivation for the generative transformation loss?
>
> Thanks for bringing up the issue. We'd like to clarify the role of generative transformation as a pivotal proxy objective that facilitates model anchoring. We've described the details in our [general response](https://openreview.net/forum?id=3mXJ9o2DNx&noteId=X8nuMF5cH6) and incorporated them in Section 3.4.
>
>
> [1] Connect, Not Collapse: Explaining Contrastive Learning for Unsupervised Domain Adaptation, ICML 2022

---

> > ### Author Response · Authors · 2023-11-23
> > **Kind Reminder**
> >
> > Dear Reviewer fbsW,
> >
> > We wish to express our sincere gratitude once again to you for the valuable contributions and considerate feedback. We would like to gently bring to your attention that the discussion phase between authors and reviewers is closing to the end (within 12 hours).
> >
> > We hope our response has addressed your concerns. Given the inclusion of further clarifications, we kindly ask for a retrospect regarding our rebuttal. Should you have any further insights to share, we are more than willing to sustain our discussion until the deadline.

---

### Official Review · Reviewer_BueW · 2023-11-01

**Soundness:** 2 fair
**Presentation:** 2 fair
**Contribution:** 2 fair
**Rating:** 3
**Confidence:** 5

**Summary:**

The paper focuses on addressing the shortcomings of self-contrastive learning in Domain Generalization (DG). To enhance domain connectivity within Contrastive Learning (CL), the authors introduce two strategies. Firstly, they suggest anchoring learned maps to pre-trained models that already exhibit the needed connectivity between training and testing domains. Secondly, they introduce a Generative Transformation Loss to further enhance alignment. The paper showcases the effectiveness of their approach, termed DCCL, through extensive experimentation on five DG benchmarks.

**Strengths:**

The majority of the paper is straightforward to understand.

**Weaknesses:**

Regarding the motivation behind the proposed method, I have three inquiries:
1. Could you elaborate on the challenges faced when implementing self-contrastive learning in DG?
2. What drove the need for more intensive data augmentation and the inclusion of cross-domain positive samples?
3. Please shed light on the rationale for model anchoring, particularly leveraging domain connectivity in pre-trained representations and its synergy with generative transformation loss.

Notably, the model is constructed on the foundation of SWAD. When evaluated across five benchmarks, the performance improvements over SWAD are recorded as 1.0, 2.9, 3.7, 0.9, and 1.0 respectively. An ablation study assessing the effectiveness of the three components of the proposed method was specifically conducted on the benchmark with a 2.9 gain. Would it be possible to also conduct the ablation study on benchmarks that achieved a performance gain of 1.0 or less? My curiosity stems from questioning whether all three components consistently contribute to the effectiveness as presented in the paper.

**Questions:**

See weaknesses above

---

> ### Author Response · Authors · 2023-11-21
> **Response to Reviewer BueW**
>
> We appreciate your helpful feedback. We wish the following answers can address your concerns.
>
> > **Q1:** Could you elaborate on the challenges faced when implementing self-contrastive learning in DG?
>
> Thanks for the question. Self-Contrastive Learning (SCL), which aligns the augmentation views of the same input, has achieved successful performance in unsupervised pre-training tasks.
>
> However, it does not naturally fit the domain generalization setting since it assumes the ability to sample $x$ from the whole data distribution; in the training stage of domain generalization, we instead are only able to access partial domains. This mismatch can lead to sub-optimal performance in domain generalization if the users mechanically adopt the classical contrastive learning loss.
>
> In Appendix C of our paper, we provide an example and detailed explanation that even attaining optimal self-contrastive learning loss cannot guarantee good performance in the domain generalization setting. To be specific, in Figure 6, we deliberately visualize the example. Slashes and spots are used to represent domains $d_1$ and $d_2$; orange and blue rectangles respectively denote classes 1 and 2. The mapping function $\varphi (\theta(x)) := (\cos (\theta), \sin(\theta))$ with $\theta(x) = (x - \mathrm{sgn}(y)) \pi$ learned on domain $d_1$ can perfectly classify the samples, and the mapping attains perfect alignment and uniformity (the objective of SCL). However, when applied to a new domain $d_2$, the classifier completely fails (0\% acc). This example highlights the potential challenges in applying self-contrastive learning effectively in the domain generalization context. We have also updated Appendix C accordingly to better elaborate on the challenges.
>
>
> > **Q2:** What drove the need for more intensive data augmentation and the inclusion of cross-domain positive samples?
>
> Building on the response to Q1, SCL in the previous example fails to obtain **intra-class connectivity** due to insufficient data augmentation and domain-separated (rather than class-separated) representations, which ultimately causes poor generalization performance.
>
> Inspired by the above analysis, we thus propose two approaches to improve domain connectivity:
> 1. applying more aggressive data augmentation and
> 2. expanding the scope of positive samples, from solely self-augmented outputs $a(x)$ to the augmentation of intra-class samples across domains.
>
> > **Q3:** Please shed light on the rationale for model anchoring, particularly leveraging domain connectivity in pre-trained representations and its synergy with generative transformation loss.
>
> The previous connectivity improvements mentioned in Q1 and Q2 only involve data transforms, while this data alignment strategy merely enhances the clustering of representations within known domains, and may not effectively bridge the gap between unseen test domains and trained domains within the same class.
>
> Therefore, we extend our focus to the model perspective. In the paper, we observed that using a pre-trained ResNet-50 model, intra-class samples from training and testing domains are scattered yet interconnected.
>
> By integrating this observation, our paper presents a unified approach, combining data and model perspectives with self-supervised objectives to bolster connectivity, thereby advancing domain generalization.
>
> Regarding generative transformation, it's a pivotal proxy objective that facilitates model anchoring. We've described the details in our [general response](https://openreview.net/forum?id=3mXJ9o2DNx&noteId=X8nuMF5cH6) and incorporated them in Section 3.4.
>
> > **Q4:** Would it be possible to also conduct the ablation study on benchmarks that achieved a performance gain of 1.0 or less?
>
> Thank you for your interest. We are glad to offer more empirical evidence to support our findings.
>
> Our previous decision to conduct the ablation study exclusively on the OfficeHome dataset was primarily to reduce computational demands. Here we additionally performed an ablation study on the VLCS dataset, as shown in the table below, where the performance gain is 0.9. These results further confirm that the three components we identified contribute consistently to the effectiveness, as detailed in our paper. We've included the experimental results in Appendix A.5 of our revision.
>
> |   Algorithm  |   C    |    L   |   S   |    V|Avg          |
> |:------------:|:---------------:|:------:|:---------------:|:---------------:|:----------------------:|
> |     SWAD     |       98.8      |  63.3  |       75.3      |       79.2      |          79.1          |
> | DCCL w/o CDC |       98.9      |  63.8  |       75.6      |       79.5      |          79.4          |
> | DCCL w/o PMA |       98.6      |  63.7  |       75.7      |       79.3      |          79.3          |
> |  DCCL w/o GT |       98.7      |  **64.3**  |       75.2      |       80.2      |          79.6          |
> | DCCL| **99.1** | 64.0 | **76.1** | **80.7** | **80.0** |

---

> > ### Author Response · Authors · 2023-11-23
> > **Kind Reminder**
> >
> > Dear Reviewer BueW,
> >
> > We wish to express our sincere gratitude once again to you for the valuable contributions and considerate feedback. We would like to gently bring to your attention that the discussion phase between authors and reviewers is closing to the end (within 12 hours).
> >
> > We hope our response has addressed your concerns. Given the inclusion of further clarifications and new experiments, we kindly ask for a retrospect regarding our rebuttal. Should you have any further insights to share, we are more than willing to sustain our discussion until the deadline.

---

### Official Review · Reviewer_ttfL · 2023-11-02

**Soundness:** 2 fair
**Presentation:** 2 fair
**Contribution:** 2 fair
**Rating:** 5
**Confidence:** 4

**Summary:**

Authors incorporate contrastive learning into domain generalization by modifying the contrastive strategy with the help of cross-domain samples and pretrained representations. Experiments on benchmarks show the effectiveness.

**Strengths:**

The idea and logic are basically complete.

Experiments show the effectiveness of the proposed algorithm, including ablation studies for three components each.

**Weaknesses:**

- The toy experiment of illustrating the drawbacks of directly applying SCL to DG is put in appendix, which is a key part of the entire logic of this paper and should be emphasized. However, this part currently seems confusing and hard to follow, and the removal of data augmentation in this experiment make it too unrealistic.
- The term "domain connectivity" is a little confusing. It is more like "domain-invariant intra-class connectivity", while "domain connectivity" seems like inner-domain connectivity.
- Experimental improvement is generally not large compared with current SOTA.

**Questions:**

- In Sec 3.2, whether samples come from the same domain or not, they belong to positive pairs as long as they are from the same class. I understand this has the benefit of not using domain labels, but if truly targeting at cross-domain positive pairs, only samples from the same class but belonging to different domains should be treated as positive pairs, otherwise it cannot be determined whether "cross-domain" is important here.
- In Sec 3.4, why is the generative transformation loss needed when there is pretrained model anchoring already? I think both of them serve as regularization to constrain the representation space not far away from the pretrained representation space. From this perspective, generative transformation loss seems a little ad-hoc.
- In appendix C, paragraphs are duplicated. As stated above, I think this part is an important component of the whole paper story, so I wonder if authors can modify it into a clearer version.

---

> ### Author Response · Authors · 2023-11-21
> **Response to Reviewer ttfL**
>
> We appreciate your helpful feedback. We hope the following answers can address your concerns.
>
> > **Q1:** The toy experiment of illustrating the drawbacks of directly applying SCL to DG is put in the appendix, which is a key part of the entire logic of this paper and should be emphasized. In Appendix C, paragraphs are duplicated. As stated above, I think this part is an important component of the whole paper story.
>
> Thank you for your suggestion. We have updated Appendix C accordingly.
>
> This example is mainly motivational (i.e., not as indispensable as parts), and thus we put it in the Appendix due to space limit. We'll incorporate this revised section into the main body of the arxiv version for coherent narrative flow.
>
> To elaborate, in Figure 6, we deliberately visualize the example. Slashes and spots are used to represent domains $d_1$ and $d_2$; orange and blue rectangles respectively denote classes 1 and 2. The mapping function $\varphi (\theta(x)) := (\cos (\theta), \sin(\theta))$ with $\theta(x) = (x - \mathrm{sgn}(y)) \pi$ learned on domain $d_1$ can perfectly classify the samples, and the mapping attains perfect alignment and uniformity (the objective of SCL). However, when applied to a new domain $d_2$, the classifier completely fails (0\% acc).
>
>
> > **Q2:** Experimental improvement is generally not large compared with the current SOTA.
>
> We hope to argue that our method's enhancement is at least on par with, if not greater than, existing baseline methods.
>
> Detailed examples: In the VLCS and OfficeHome datasets, our methods – DCCL, SelfReg, PCL, and MIRO – show improvements over the state-of-the-art (SOTA) benchmarks by margins of 0.4/1.1, 0.3/-0.8, -1.0/1.0, and 0.5/0.8, respectively. Furthermore, within the domain generalization community,  with the same backbone, fixed data splitting, and long-term development, it's important to note that achieving significant absolute improvements is challenging. It often requires the combined efforts of multiple research papers.
>
> > **Q3:** The term "domain connectivity" is a little confusing. It is more like "domain-invariant intra-class connectivity", while "domain connectivity" seems like inner-domain connectivity.
>
> Thanks for the advice. We follow the usage in [1] and have updated the term 'domain connectivity' to 'intra-class connectivity' for greater clarity and to more accurately convey our intent.
>
> [1] Connect, Not Collapse: Explaining Contrastive Learning for Unsupervised Domain Adaptation, ICML 2022
>
> > **Q4:** Question regarding whether samples come from the same domain or not, they belong to positive pairs as long as they are from the same class.
>
>
> We clarify that the 'cross-domain' positive samples in our study encompass both samples from the same and different domains.
>
> This is because we lack domain labels for the samples. In Appendix A.5 of our submitted paper, we provide a rationale for using cross-domain contrast (CDC), which shows an accuracy of 71.8\% as noted in Table 2. To contrast, we conducted a baseline experiment using only within-domain positive samples and observed a substantial drop in accuracy when compared to CDC (from 71.8\% to 70.4\%). Furthermore, an oracle experiment exclusively using cross-domain positive pairs yielded similar results (from 71.8\% to 71.9\%). These results validate our approach and suggest that, if domain information is available, its effective utilization might require thoughtful design to achieve notable improvements.
>
> > **Q5:** why is the generative transformation loss needed when there is pretrained model anchoring already?
>
> Thanks for bringing up the issue. We'd like to clarify the role of generative transformation as a pivotal proxy objective that facilitates model anchoring. We've described the details in our [general response](https://openreview.net/forum?id=3mXJ9o2DNx&noteId=X8nuMF5cH6) and incorporated them in Section 3.4.

---

> > ### Comment · Reviewer_ttfL · 2023-11-22
> >
> > Thanks for your reply! Many of my concerns have been addressed, and I appreciate the efforts authors have made in their rebuttal and paper revision. Generally I like the story and logic of this paper, but the experimental part is still not that convincing to me. I think there is still some space for improvements in the current version.
> >
> > In summary, I decide to raise my score from 3 to 5.

---

> > > ### Author Response · Authors · 2023-11-22
> > >
> > > Thank you for your constructive feedback! We're glad that we've addressed many of your concerns. Regarding the experimental section, our findings demonstrate a notable improvement over the baselines, substantiated by significance testing. Your acknowledgment through an increased score is greatly appreciated, and we remain dedicated to continuous enhancements.

---

### Official Review · Reviewer_NxUh · 2023-11-06

**Soundness:** 3 good
**Presentation:** 3 good
**Contribution:** 2 fair
**Rating:** 5
**Confidence:** 3

**Summary:**

The paper proposes a new domain generalization method that consists of a ERM loss, a contrastive learning loss with cross-domain data augmentation and a generative transformation loss that exploits the supervised signal at the inter-sample. Experimental results seem to validate the effectiveness of the proposed method.

**Strengths:**

- The paper is clearly written and generally easy to follow.

- The experimental results look good and the proposed DCCL achieves very competitive performance on a few benchmarks.

**Weaknesses:**

- The combination of contrastive learning and continual learning has been explored and shown effective multiple times. See [1,2]. The contribution of this work is mostly on how you augment the positive samples. The way the paper augments the samples across different domains is similar to [3,4].

- I fail to understand why generative transformation helps. Could the authors elaborate more on why learning an additional decoder for the generative transformation can help domain generalization? I can see that this is to exploit more information from the sample itself, but why it can make domain generalization better is unclear.

- Overall, I find the proposed method a bit ad-hoc, as it combines multiple disconnected components to improve the final performance. Why these components can work coherently to benefit the domain generalization task is unclear and also poorly motivated. It is insufficient to simply state that some losses are for intra-sample level and some for inter-sample level. It still does not explain why it works.

[1] SelfReg: Self-supervised Contrastive Regularization for Domain Generalization. ICCV 2021

[2] PCL: Proxy-based Contrastive Learning for Domain Generalization. CVPR 2022

[3] Towards Principled Disentanglement for Domain Generalization, CVPR 2022

[4] Model-based domain generalization, NeurIPS 2021

**Questions:**

See the weakness section.

---

> ### Author Response · Authors · 2023-11-21
> **Response to Reviewer NxUh**
>
> We appreciate your helpful feedback. We hope the following answers can address your concerns.
>
> > **Q1:** The combination of contrastive learning and continual learning has been explored and shown effective multiple times.
>
> Thanks for raising the question.
>
> Previous research, as referenced in [1,2], primarily tackled the positive data alignment issue in domain generalization by developing proxy objectives. Meanwhile, studies in [3,4] advanced the alignment of training domains, utilizing specially designed image generators to create domain-transformed positive samples.
>
> However, these approaches primarily involve data transforms. In contrast, our work diverges by exhaustively exploring the concept of connectivity from both data and model perspectives.
> On one hand (data), we build upon the shortcomings of traditional methods that align augmentation views of the same input, as explained in Appendix C.
> We propose two direct improvements for enhancing domain connectivity through data:
> 1. implementing more aggressive data augmentation, and
> 2. broadening the range of positive samples, extending beyond self-augmented outputs to include intra-class samples across different domains.
> Nevertheless, this data alignment strategy merely enhances the clustering of representations within known domains, and may not effectively bridge the gap between unseen test domains and trained domains within the same class.
>
> We therefore extend our focus to the model perspective. In the paper, we observed that using a pre-trained ResNet-50 model, intra-class samples from training and testing domains are scattered yet interconnected. By integrating this observation, our paper presents a unified approach, combining data and model perspectives with self-supervised objectives to bolster connectivity, thereby advancing domain generalization.
>
> > **Q2:** Could the authors elaborate more on why learning an additional decoder for the generative transformation can help domain generalization?
>
> Thanks for bringing up the issue. We'd like to clarify the role of generative transformation as a pivotal proxy objective that facilitates model anchoring. This can be understood from multiple angles:
> 1. Echoing the findings in [2] that directly aligning positive pairs across vastly different domains often results in poor performance, our research similarly identifies a substantial gap in the representations of pre-trained and fine-tuned models. Direct alignment using contrastive learning, as evidenced by our findings in Table 2, tends to be sub-optimal. In response, we introduce the concept of variational generative loss to **comprehend the transformation process** and bridge these representational gaps.
> 2. Additionally, the generative transformation module is designed to reconstruct the features of the pre-trained model **at an intra-sample level**. This complements the inter-sample level supervision provided by contrastive loss. The module, along with its associated loss function, is intended to provide a more enriched supervised signal, encapsulating crucial within-sample information. This, in turn, supports and enhances the anchoring of the pre-trained model.
>
> > **Q3:** Why these components can work coherently to benefit the domain generalization task is unclear.
>
> As shown in the response to Q1, our approach combines data and model perspectives with self-supervised objectives to bolster connectivity, thereby advancing domain generalization.
>
>
> [1] SelfReg: Self-supervised Contrastive Regularization for Domain Generalization. ICCV 2021
>
> [2] PCL: Proxy-based Contrastive Learning for Domain Generalization. CVPR 2022
>
> [3] Towards Principled Disentanglement for Domain Generalization, CVPR 2022
>
> [4] Model-based domain generalization, NeurIPS 2021

---

> > ### Author Response · Authors · 2023-11-23
> > **Kind Reminder**
> >
> > Dear Reviewer NxUh,
> >
> > We wish to express our sincere gratitude once again to you for the valuable contributions and considerate feedback. We would like to gently bring to your attention that the discussion phase between authors and reviewers is closing to the end (within 12 hours).
> >
> > We hope our response has addressed your concerns. Given the inclusion of further clarifications, we kindly ask for a retrospect regarding our rebuttal. Should you have any further insights to share, we are more than willing to sustain our discussion until the deadline.

---

### Public Comment · ~Boris_Nico1 · 2023-11-13

Hi, authors. DN$^2$A [1] (which has already been accepted by CVPR 2023) already proposed to utilize the "connectivity" concept to analyze the failure of self-supervised contrastive learning in DG, and accordingly propose stronger data augmentation and cross domain nearest neighbors as positive samples to improve the performance of self-supervised contrastive learning in DG. I thinks this paper should cite this very related reference for clarifying the difference and novelty. Thanks.

[1] Promoting Semantic Connectivity: Dual Nearest Neighbors Contrastive Learning for Unsupervised Domain Generalization. CVPR2023.

---

> ### Author Response · Authors · 2023-11-21
>
> We appreciate the suggested [reference](https://openreview.net/forum?id=Iewi8zwGsZr). We've cited and discussed it in the Appendix B.1 of our paper. To help differentiate our contributions from this paper, we summarize the main points thereof as follows. As the mathematical notations is non-standard in [1] (such as the expectation of distribution in Def. 1 and minimum product $\min x_i^+ x_j^+$ of two augmented samples in Eqn. (14)), there might be misunderstanding.
>
> - The concepts of "connectivity" differ in two papers.
>
> [2] motivates both this project and [1] (as claimed in Section 3.2 of [1]). In [2], the original definition of intra-class connectivity appears in Assumption 4.5 just means the augmentation graph (c.f. their Def. 4.4) is connected. We follow the same definition of "connectivity" in our paper.
>
> However, [1] is based on the concepts of intra-domain, intra-class, and semantic connectivity as
>
> > Definition 1. (Semantic Connectivity) For any input $x \in \mathcal{X}$, let $\mathcal{A}(\cdot \mid x)$ be the distribution of its augmentations $A$. Let $C$ be the joint distribution on $\mathcal{X} \times \mathcal{X}$ of augmented views of images $x_i, x_j$ as $C\left(x_i^{+}, x_j^{+}\right)=\mathcal{A}\left(x_i^{+} \mid x_i\right) \mathcal{A}\left(x_j^{+} \mid x_j\right)$. Then we have intra-domain $C_\alpha$ and intra-class $C_\beta$ connectivity.
> >
> > $$
> C_\alpha:=\\mathbb{E}\_{d \sim P_D^S} \\mathbb{E}\_{x_i, x_j \sim P\_d^{S\_{\\mathrm{UL}}}} C\left(x\_i^{+}, x\_j^{+}\right), \quad C_\beta:=\\mathbb{E}\_{y \sim P\_Y^S} \\mathbb{E}\_{x_i, x_j \sim P_y^{S\_{\\mathrm{UL}}}} C\left(x_i^{+}, x_j^{+}\right)
> $$
> > Then, the semantic connectivity is defined as $C_s:=C_\beta / C_\alpha$.
>
> For the reader's convenience, we remark their definition of the joint distribution $C()$ is uncommon: it is a constant multiple of the so-called minimum product (in their Eqn. (14-16))
>
> $$
> e_A\left(x_i^{+}, x_j^{+}\right)=\min _{x_i^{+} \in A\left(x_i\right), x_j^{+} \in A\left(x_j\right)} x_i^{+} x_j^{+}.
> $$
> We argue it is obvious that the connectivity concept based on the joint distribution in their paper is quite different from ours and [2]; therefore the theoretical motivation behind two papers are indeed different.
>
> - The methods proposed are different.
>
> Except for the common strategy of strong augmentation recommended by the contrastive learning theory paper [2], our proposed methods are different from the ones in [1]. They propose two nearest-neighbor-based methods for constructing positive pairs, while our main contribution lies in the exploitation of both the pre-trained models and the intra-class data connectivity.
>
> ---
>
> [1] Promoting Semantic Connectivity: Dual Nearest Neighbors Contrastive Learning for Unsupervised Domain Generalization. CVPR 2023.
>
> [2] Chaos is a Ladder: A New Theoretical Understanding of Contrastive Learning via Augmentation Overlap. ICLR 2022.

---

### Author Response · Authors · 2023-11-21
**General Response**

We thank all the reviewers for their important feedback on our work. We are confident that any concerns raised are fully addressed and are mostly due to misunderstandings. We will revise the manuscript according to the provided feedback. It is more than welcome if the reviewers consider open questions remain and hope to engage with us.


Regarding the concerns raised, we highlight the response to common concerns here and have addressed each specific concern individually to each reviewer.

> The novelty of our proposed domain-connecting contrastive learning (DCCL).

It's important to note that our paper's central theme is not just introducing the concept of connectivity and theoretical analysis, but rather emphasizing connectivity enforcement **from both data and model perspectives**.

> The rationale of designing generative transformation.

For the design of generative transformation, we'd like to clarify its role as a pivotal proxy objective that facilitates model anchoring. This can be understood from multiple angles: (i) Echoing the findings in [2], which point out that directly aligning positive pairs across vastly different domains often results in poor performance, our research similarly identifies a substantial gap in the representations of pre-trained and fine-tuned models. Direct alignment using contrastive learning, as evidenced by our findings in Table 2, tends to be sub-optimal. In response, we introduce the concept of variational generative loss to comprehend the transformation process and bridge these representational gaps. (ii) Additionally, the generative transformation module is designed to reconstruct the features of the pre-trained model at an intra-sample level. This complements the inter-sample level supervision provided by contrastive loss. The module, along with its associated loss function, is intended to provide a more enriched supervised signal, encapsulating crucial within-sample information. This, in turn, supports and enhances the anchoring of the pre-trained model. We've incorporated the rationale of generative transformation in Section 3.4 of the revision.

> Additional Ablation Study to verify the effectiveness of the proposed modules.

We also added further ablation study (as requested) on the VLCS dataset and have included these results in Appendix A.5.

|   Algorithm  |        C        |    L   |        S        |        V        |           Avg          |
|:------------:|:---------------:|:------:|:---------------:|:---------------:|:----------------------:|
|     SWAD     |       98.8      |  63.3  |       75.3      |       79.2      |          79.1          |
| DCCL w/o CDC |       98.9      |  63.8  |       75.6      |       79.5      |          79.4          |
| DCCL w/o PMA |       98.6      |  63.7  |       75.7      |       79.3      |          79.3          |
|  DCCL w/o GT |       98.7      |  **64.3**  |       75.2      |       80.2      |          79.6          |
|     DCCL     | **99.1** | 64.0 | **76.1** | **80.7** | **80.0** |


---

Last but not least, we also want to thank the Reviewers for noting the strengths of our paper, namely:

- Clearly written and easy to follow. (Reviewer NxUh, ttfL, BueW, 2QYf)
- Competitive performance on the domain generalization benchmarks. (Reviewer NxUh, ttfL, 2QYf)
- Interesting application of CL to DG, the proposed idea is very simple yet effective. (Reviewer fbsW)
- Clear and reasonable definition of domain connectivity, enhancing our understanding of this important aspect. Noteworthy contribution of enforcing connectivity for DG. (Reviewer 2QYf)

---

### Meta-Review · Area_Chair_JfA7 · 2023-12-04

**Metareview:**

The paper studies the application of contrastive learning methods to improve performance on domain generalization. The direction is interesting but lacks rigor. From the draft, it is not fully clear why it deteriorates performance, the paper rather provides an intuition thereof with an example. Overall, the idea of incorporating similarity among domains was not found to be novel and the experiments were not sufficiently convincing.

**Justification For Why Not Higher Score:**

There is an unclear theoretical contribution, it rather shows an example showing that it can deteriorate performance but overstates the claim. Experiments were found to be marginal.

**Justification For Why Not Lower Score:**

N/A

---

### Decision · Program_Chairs · 2024-01-16

Reject